# Adaptive Stochastic Gradient Algorithm for Black-box Multi-Objective Learning

**Feiyang Ye**[1,2,*]**, Yueming Lyu**[3,4,*]**, Xuehao Wang**[1]**, Yu Zhang**[1,5,†]**, Ivor W. Tsang**[2,3,4,6]

[1]Department of Computer Science and Engineering, Southern University of Science and Technology
[2]Australian Artificial Intelligence Institute, University of Technology Sydney
[3]Centre for Frontier AI Research, Agency for Science, Technology and Research, Singapore
[4]Institute of High Performance Computing, Agency for Science, Technology and Research, Singapore
[5]Shanghai Artificial Intelligence Laboratory
[6]School of Computer Science and Engineering, Nanyang Technological University
{feiyang.ye.uts,xuehaowangfi,yu.zhang.ust}@gmail.com
{Lyu_Yueming,Ivor_Tsang}@cfar.a-star.edu.sg

## Abstract

Multi-objective optimization (MOO) has become an influential framework for various machine learning problems, including reinforcement learning and multi-task learning. In this paper, we study the black-box multi-objective optimization problem, where we aim to optimize multiple potentially conflicting objectives with function queries only. To address this challenging problem and find a Pareto optimal solution or the Pareto stationary solution, we propose a novel adaptive stochastic gradient algorithm for black-box MOO, called ASMG. Specifically, we use the stochastic gradient approximation method to obtain the gradient for the distribution parameters of the Gaussian smoothed MOO with function queries only. Subsequently, an adaptive weight is employed to aggregate all stochastic gradients to optimize all objective functions effectively. Theoretically, we explicitly provide the connection between the original MOO problem and the corresponding Gaussian smoothed MOO problem and prove the convergence rate for the proposed ASMG algorithm in both convex and non-convex scenarios. Empirically, the proposed ASMG method achieves competitive performance on multiple numerical benchmark problems. Additionally, the state-of-the-art performance on the black-box multi-task learning problem demonstrates the effectiveness of the proposed ASMG method.

## 1 Introduction

Multi-objective optimization (MOO) involves optimizing multiple potentially conflicting objectives simultaneously (Deb et al., 2016; Fliege & Svaiter, 2000). In recent years, MOO has drawn intensive attention in a wide range of applications, including meta-learning (Ye et al., 2021; Yu et al., 2022), reinforcement learning (Thomas et al., 2021; Prabhakar et al., 2022), learning-to-rank (LTR) problems Mahapatra et al. (2023a;b), and multi-task learning (Momma et al., 2022; Fernando et al., 2022; Zhou et al., 2022b; Lin et al., 2022; 2023; Ye et al., 2024). A typical MOO problem is formulated as

$$\min_{\boldsymbol{x} \in \mathcal{X}} F(\boldsymbol{x}) := (F_1(\boldsymbol{x}), F_2(\boldsymbol{x}), \ldots, F_m(\boldsymbol{x})), \tag{1}$$

where $m \geq 2$ denotes the number of objectives, $\mathcal{X} \subseteq \mathbb{R}^d$ and $d$ represents the parameter dimension. The objective function $F_i : \mathbb{R}^d \to \mathbb{R}$ satisfies $F_i(\boldsymbol{x}) \geq -\infty$ for $i = 1, \ldots, m$.

Solving the MOO problem is challenging because in most cases it cannot find a common parameter that minimizes all objective functions simultaneously. Therefore, a widely adopted strategy is to find a Pareto optimal solution or a Pareto stationary solution. To achieve this goal, a typical gradient-based method is the multiple gradient descent algorithm (MGDA) (Désidéri, 2012). The basic idea of

---

[*]Equal contribution.
[†]Corresponding author.

MGDA is to iteratively update the variable $\boldsymbol{x}$ via a common descent direction for all the objectives through a convex combination of gradients from individual objectives. Various MGDA-based MOO algorithms (Yu et al., 2020; Liu et al., 2021; Fernando et al., 2022; Zhou et al., 2022b) have been proposed to adjust the multiple gradients to seek a common descent direction that simultaneously decreases all the objectives. Those gradient-based MOO algorithms have been successfully applied in a wide range of applications, especially for multi-task learning (MTL) (Sener & Koltun, 2018; Zhang & Yang, 2022).

However, in many MOO-based learning problems, the gradient of the objective $F(\boldsymbol{x})$ w.r.t. the variable $\boldsymbol{x}$ cannot be explicitly calculated, making problem (1) a black-box MOO problem (Wang & Shan, 2004; Žilinskas, 2014). For instance, many large models such as large language models (LLMs) (Devlin et al., 2018; Raffel et al., 2020; Yu et al., 2023) are released in the service and are only allowed for access with APIs (Brown et al., 2020). In such scenarios, users can only query the large models without accessing gradients to accomplish tasks of interest (Sun et al., 2022b;a), and gradient-based MOO methods are no longer applicable since they all rely on the availability of true gradients or stochastic gradients w.r.t. the variable $\boldsymbol{x}$. Several kinds of approaches have been widely studied for black-box MOO, such as Bayesian optimization (BO) (Konakovic Lukovic et al., 2020; Zhang & Golovin, 2020) and genetic algorithms (GA) (Laumanns et al., 2002; Wang & Shan, 2004; Chen et al., 2012; Arrieta et al., 2018). Among those methods, BO methods are good at dealing with low-dimensional expensive black-box MOO problems, while GA is to explore the entire Pareto optimal set, which is computationally expensive for machine learning problems, and usually lacks convergence analysis. Therefore, those limitations motivate us to design an algorithm for black-box MOO that can effectively reach a Pareto optimal solution or a Pareto stationary solution for relatively high-dimensional learning problems with affordable evaluations and convergence guarantee.

To achieve that, in this paper, we propose a novel Adaptive Stochastic Multi-objective Gradient (ASMG) algorithm for black-box MOO by taking advantage of gradient-based MOO methods. Specifically, the ASMG method first smoothes each objective to their expectation over a Gaussian distribution, leading to Gaussian smoothed objectives. Then it iteratively updates the parameterized distribution via a common search direction aggregated by the approximated stochastic gradients for all smoothed objectives. We explore the connections between the MOO and the corresponding Gaussian smoothed MOO and provide a convergence analysis for the proposed ASMG algorithm under both convex and non-convex scenarios. Moreover, experiments on various numerical benchmark problems and a black-box multi-task learning problem demonstrate the effectiveness of the proposed ASMG method.

The main contributions of this work are three-fold: (i) We propose a novel ASMG algorithm for black-box multi-objective optimization. To the best of our knowledge, we are the first to design a stochastic gradient algorithm for black-box MOO with a theoretical convergence guarantee. (ii) Theoretically, we explicitly provide the connection of the Pareto optimal and stationary conditions between the original MOO and the corresponding Gaussian-smoothed MOO. Moreover, we prove the convergence rate for the proposed ASMG algorithm in both convex and non-convex cases. (iii) Empirically, the proposed ASMG algorithm achieves competitive performances on multiple numerical benchmark problems. Moreover, the state-of-the-art performance on black-box multi-task learning problems demonstrates the effectiveness of the proposed ASMG method.

**Notation and Symbols.** $\|\cdot\|_1$, $\|\cdot\|_2$, and $\|\cdot\|_\infty$ denote the $l_1$ norm, $l_2$ norm, and $l_\infty$ norm for vectors, respectively. $\|\cdot\|_F$ denote the Frobenius norm for matrices. $\Delta^m$ denotes an $m$-dimensional simplex. $\mathcal{S}^+$ denotes the set of positive semi-definite matrices. $\frac{X}{Y}$ denotes the elementwise division operation when $X$ and $Y$ are vectors, and the elementwise division operation for diagonal elements in $X$ and $Y$ when they are diagonal matrices. For a square matrix $\mathbf{X}$, $\mathrm{diag}(\mathbf{X})$ is a vector with diagonal entries in $\mathbf{X}$, and if $\mathbf{x}$ is a vector, $\mathrm{diag}(\mathbf{x})$ is a diagonal matrix with $\mathbf{x}$ as its diagonal entries. Define $\|X\|_Y := \sqrt{\langle X, YX \rangle}$ for a matrix $Y \in \mathcal{S}^+$ or a non-negative vector $Y$, where $\langle \cdot, \cdot \rangle$ denotes the inner product under the $l_2$ norm for vectors and the inner product under the Frobenius norm for matrices.

## 2 BACKGROUND

In this section, we introduce useful concepts of MOO and stochastic gradient approximation strategies for black-box optimization.

## 2.1 MULTI-OBJECTIVE OPTIMIZATION

In MOO (Deb et al., 2016), we are interested in finding solutions that can not be improved simultaneously for all the objectives, leading to the notion of Pareto optimality, which is defined as follows for the problem (1).

**Definition 2.1.** For any two point $\boldsymbol{x}_1, \boldsymbol{x}_2 \in \mathcal{X}$, we define that $\boldsymbol{x}_1$ dominates $\boldsymbol{x}_2$ if $F_i(\boldsymbol{x}_1) \leq F_i(\boldsymbol{x}_2)$ holds for $i = 1, \ldots, m$, and $F_i(\boldsymbol{x}_1) \neq F_i(\boldsymbol{x}_2)$ holds for some $i$. A point $\boldsymbol{x}^* \in \mathcal{X}$ is called Pareto optimal if it is not dominated by any other point in $\mathcal{X}$. The set of all Pareto optimal solutions forms the Pareto set. The set of objective values $F(\boldsymbol{x}^*)$ for all the Pareto optimal is called the Pareto front.

We then present the sufficient condition for determining Pareto optimal.

**Proposition 2.2.** *For MOO problem (1), if all objective $F_i(\boldsymbol{x})$ for $i = 1, \ldots, m$ are convex functions and there exists $\boldsymbol{\lambda} \in \Delta^{m-1}$ such that $\boldsymbol{x}^* = \arg\min_{\boldsymbol{x}} \boldsymbol{\lambda}^\top F(\boldsymbol{x})$, then $\boldsymbol{x}^*$ is a Pareto optimal.*

The above proposition implies that the minimizer of any linearization is Pareto optimal (Zhou et al., 2022b). In the general nonconvex cases, MOO aims to find the Pareto stationary solution (Fliege et al., 2019). If a point $\hat{\boldsymbol{x}}$ is a Pareto stationary solution, then there is no common descent direction for all $F_i(\boldsymbol{x})$'s ($i = 1, \ldots, m$) at $\hat{\boldsymbol{x}}$. For the Pareto stationary condition, we have the following proposition according to Proposition 1 in Zhou et al. (2022b).

**Proposition 2.3.** *For MOO problem (1), (i) we say $\boldsymbol{x}^* \in \mathcal{X}$ is a Pareto stationary solution if there exist $\boldsymbol{\lambda} \in \Delta^{m-1}$ such that $\|\sum_{i=1}^m \lambda_i \nabla F_i(\boldsymbol{x}^*)\| = 0$; (ii) we say $\boldsymbol{x}^* \in \mathcal{X}$ is a $\epsilon$-accurate Pareto stationary solution if $\min_{\boldsymbol{\lambda}} \|\sum_{i=1}^m \lambda_i \nabla F_i(\boldsymbol{x}^*)\|^2 \leq \epsilon$ where $\boldsymbol{\lambda} \in \Delta^{m-1}$.*

## 2.2 STOCHASTIC GRADIENT APPROXIMATION STRATEGIES

Inspired by evolution strategy, the stochastic gradient approximation method (Wierstra et al., 2014; Lyu & Tsang, 2021) for black-box optimization, instead of maintaining a population of searching points, iteratively updates a search distribution by stochastic gradient approximation.

The stochastic gradient approximation strategies employed in black-box optimization typically follow a general procedure. Firstly, a parameterized search distribution is utilized to generate a batch of sample points. Then the sample points allow the algorithm to capture the local structure of the fitness function and appropriately estimate the stochastic gradient to update the distribution. Specifically, when $\boldsymbol{\theta}$ denotes the parameters of the search distribution $p_{\boldsymbol{\theta}}(\boldsymbol{x})$ and $f(\boldsymbol{x})$ denotes a single objective function for sample $\boldsymbol{x}$, the expected fitness under the search distribution can be defined as $J(\boldsymbol{\theta}) = \mathbb{E}_{p_{\boldsymbol{\theta}}(\boldsymbol{x})}[f(\boldsymbol{x})]$. Based on this definition, we can obtain the Monte Carlo estimate of the search gradient as

$$\nabla_{\boldsymbol{\theta}} J(\boldsymbol{\theta}) = \frac{1}{N} \sum_{j=1}^{N} f(\boldsymbol{x}_j) \nabla_{\boldsymbol{\theta}} \log p_{\boldsymbol{\theta}}(\boldsymbol{x}_j), \tag{2}$$

where $N$ denotes the number of samples, and $\boldsymbol{x}_j$ denotes the $j$-th sample. Therefore, the stochastic gradient $\nabla_{\boldsymbol{\theta}} J(\boldsymbol{\theta})$ provides a search direction in the space of search distributions.

## 3 METHODOLOGY

In this section, we introduce the proposed ASMG algorithm. Firstly, we formulate the black-box MOO as a min-max optimization problem and solve it in Section 3.1. Then in Section 3.2, we derive the update formula of parameters in the search distribution under the Gaussian sampling.

## 3.1 BLACK-BOX MULTI-OBJECTIVE OPTIMIZATION

We aim to minimize the MOO problem (1) with only function queries. Due to the lack of gradient information in black-box optimization, we use the stochastic gradient approximation method. Specifically, the objective of the original MOO is smoothed to the expectation of $F(\boldsymbol{x})$ under a parametric search distribution $p_{\boldsymbol{\theta}}(\boldsymbol{x})$ with parameter $\boldsymbol{\theta}$, i.e., $J_i(\boldsymbol{\theta}) = \mathbb{E}_{p_{\boldsymbol{\theta}}(\boldsymbol{x})}[F_i(\boldsymbol{x})]$ for $i = 1, \ldots, m$. Then the optimal parameter $\boldsymbol{\theta}$ is found by minimizing the following smoothed MOO problem as

$$\min_{\boldsymbol{\theta}} J(\boldsymbol{\theta}) := (J_1(\boldsymbol{\theta}), J_2(\boldsymbol{\theta}), \ldots, J_m(\boldsymbol{\theta})). \tag{3}$$

By following Wierstra et al. (2014); Lyu & Tsang (2021), the search distribution is assumed to be a Gaussian distribution, i.e., $p_{\boldsymbol{\theta}}(\boldsymbol{x}) = \mathcal{N}(\boldsymbol{x} \mid \boldsymbol{\mu}, \boldsymbol{\Sigma})$ where $\boldsymbol{\mu}$ denotes the mean and $\boldsymbol{\Sigma}$ denotes the covariance matrix, and correspondingly $\boldsymbol{\theta}$ includes $\boldsymbol{\mu}$ and $\boldsymbol{\Sigma}$, i.e., $\boldsymbol{\theta} = \{\boldsymbol{\mu}, \boldsymbol{\Sigma}\}$. We denote $J_i(\boldsymbol{\theta}) = J_i(\boldsymbol{\mu}, \boldsymbol{\Sigma})$ for $i = 1, \ldots, m$. This Gaussian-smoothed MOO method can effectively estimate stochastic gradients, and enable a more accurate search direction for the distribution to address high-dimensional black-box MOO problems. The connection between this Gaussian smoothed MOO problem (3) and problem (1) is shown in Section 4.

Here we aim to derive an update formulation for $\boldsymbol{\theta}$. To optimize all the Gaussian smoothed objective functions effectively, inspired by MGDA, we can find a parameter $\boldsymbol{\theta}$ to maximize the minimum decrease across all smoothed objectives in each iteration as

$$\max_{\boldsymbol{\theta}} \min_{i \in [m]} (J_i(\boldsymbol{\theta}_t) - J_i(\boldsymbol{\theta})) \approx \max_{\boldsymbol{\theta}} \min_{i \in [m]} \langle \nabla_{\boldsymbol{\theta}} J_i(\boldsymbol{\theta}_t), \boldsymbol{\theta}_t - \boldsymbol{\theta} \rangle, \tag{4}$$

where $\nabla_{\boldsymbol{\theta}} J_i(\boldsymbol{\theta}_t) = \nabla_{\boldsymbol{\theta}} \mathbb{E}_{p_{\boldsymbol{\theta}_t}}[F_i(\boldsymbol{x})]$ denotes the derivative of $J_i(\boldsymbol{\theta})$ w.r.t. $\boldsymbol{\theta} = \{\boldsymbol{\mu}, \boldsymbol{\Sigma}\}$ at $\boldsymbol{\theta}_t = \{\boldsymbol{\mu}_t, \boldsymbol{\Sigma}_t\}$. In Eq. (4), the first-order Taylor expansion is used to derive the approximation with an assumption that the variable $\boldsymbol{\theta}$ is close to $\boldsymbol{\theta}_t$. To further make this assumption hold, we add a regularization term to maximize $-\frac{1}{\beta_t} \mathrm{KL}(p_{\boldsymbol{\theta}} \| p_{\boldsymbol{\theta}_t})$ into Eq. (4), and then the objective function to update $\boldsymbol{\theta}$ is formulated as

$$\min_{\boldsymbol{\theta}} \max_{\boldsymbol{\lambda}^t \in \Delta^{m-1}} \Big\langle \sum_{i=1}^m \lambda_i^t \nabla_{\boldsymbol{\theta}} J_i(\boldsymbol{\theta}_t), \boldsymbol{\theta} - \boldsymbol{\theta}_t \Big\rangle + \frac{1}{\beta_t} \mathrm{KL}(p_{\boldsymbol{\theta}} \| p_{\boldsymbol{\theta}_t}). \tag{5}$$

Note that problem (5) is convex w.r.t. $\boldsymbol{\theta}$ and concave w.r.t. $\boldsymbol{\lambda}$ for Gaussian distribution $p_{\boldsymbol{\theta}}$. Then using Von Neumann-Fan minimax theorem (Borwein, 2016), we can switch the order of the min and max operators, leading to an equivalent problem as

$$\max_{\boldsymbol{\lambda}^t \in \Delta^{m-1}} \min_{\boldsymbol{\theta}} \Big\langle \sum_{i=1}^m \lambda_i^t \nabla_{\boldsymbol{\theta}} J_i(\boldsymbol{\theta}_t), \boldsymbol{\theta} - \boldsymbol{\theta}_t \Big\rangle + \frac{1}{\beta_t} \mathrm{KL}(p_{\boldsymbol{\theta}} \| p_{\boldsymbol{\theta}_t}). \tag{6}$$

By solving the inner problem of problem (6), we can obtain the update formulations for $\boldsymbol{\mu}$ and $\boldsymbol{\Sigma}$ in the $t$-th iteration as

$$\boldsymbol{\mu}_{t+1} = \boldsymbol{\mu}_t - \beta_t \boldsymbol{\Sigma}_t \sum_{i=1}^m \lambda_i^t \nabla_{\boldsymbol{\mu}} J_i(\boldsymbol{\theta}_t), \quad \boldsymbol{\Sigma}_{t+1}^{-1} = \boldsymbol{\Sigma}_t^{-1} + 2\beta_t \sum_{i=1}^m \lambda_i^t \nabla_{\boldsymbol{\Sigma}} J_i(\boldsymbol{\theta}_t), \tag{7}$$

where $\nabla_{\boldsymbol{\mu}} J_i(\boldsymbol{\theta}_t)$ and $\nabla_{\boldsymbol{\Sigma}} J_i(\boldsymbol{\theta}_t)$ denote the derivative of $J_i(\boldsymbol{\theta})$ w.r.t. $\boldsymbol{\mu}$ and $\boldsymbol{\Sigma}$ at $\boldsymbol{\mu} = \boldsymbol{\mu}_t$ and $\boldsymbol{\Sigma} = \boldsymbol{\Sigma}_t$, respectively. To obtain those two gradients, in the following theorem, we prove that we only need function queries.

**Theorem 3.1.** *(Wierstra et al., 2014) The gradient of the expectation of an integrable function $F_i(\boldsymbol{x})$ under a Gaussian distribution $p_{\boldsymbol{\theta}} := \mathcal{N}(\boldsymbol{\mu}, \boldsymbol{\Sigma})$ with respect to the mean $\boldsymbol{\mu}$ and the covariance $\boldsymbol{\Sigma}$ can be expressed as*

$$\nabla_{\boldsymbol{\mu}} \mathbb{E}_{p_{\boldsymbol{\theta}}}[F_i(\boldsymbol{x})] = \mathbb{E}_{p_{\boldsymbol{\theta}}}[\boldsymbol{\Sigma}^{-1}(\boldsymbol{x} - \boldsymbol{\mu})F_i(\boldsymbol{x})], \tag{8}$$

$$\nabla_{\boldsymbol{\Sigma}} \mathbb{E}_{p_{\boldsymbol{\theta}}}[F_i(\boldsymbol{x})] = \frac{1}{2} \mathbb{E}_{p_{\boldsymbol{\theta}}}\big[\big(\boldsymbol{\Sigma}^{-1}(\boldsymbol{x} - \boldsymbol{\mu})(\boldsymbol{x} - \boldsymbol{\mu})^\top \boldsymbol{\Sigma}^{-1} - \boldsymbol{\Sigma}^{-1}\big)F_i(\boldsymbol{x})\big]. \tag{9}$$

According to Theorem 3.1, to calculate the gradients, we need to calculate the inverse covariance matrix, which is computationally expensive in high dimensions, and hence to reduce the computational cost, we assume that the covariance matrix $\boldsymbol{\Sigma}$ is a diagonal matrix.

Then substituting Eq. (7) into problem (6), it can be approximated by the following quadratic programming (QP) problem as

$$\min_{\boldsymbol{\lambda}^t \in \Delta^{m-1}} \Big\| \sum_{i=1}^m \lambda_i^t \boldsymbol{p}_i^t \Big\|^2 + 2 \Big\| \sum_{i=1}^m \lambda_i^t \boldsymbol{h}_i^t \Big\|^2, \tag{10}$$

where $\boldsymbol{p}_i^t = \boldsymbol{\Sigma}_t^{\frac{1}{2}} \nabla_{\boldsymbol{\mu}} J_i(\boldsymbol{\theta}_t)$ and $\boldsymbol{h}_i^t = \mathrm{diag}(\boldsymbol{\Sigma}_t \nabla_{\boldsymbol{\Sigma}} J_i(\boldsymbol{\theta}_t))$. The detailed derivation is put in the Appendix A. Problem (10) is obviously convex and the objective function of problem (10) can be simplified to $\boldsymbol{\lambda}^{t\top} \Lambda^\top \Lambda \boldsymbol{\lambda}^t$, where $\boldsymbol{\lambda}^t = (\lambda_1^t, \ldots, \lambda_m^t)^\top$ and $\Lambda = (((\boldsymbol{p}_i^t)^\top, \sqrt{2}(\boldsymbol{h}_i^t)^\top)^\top, \ldots, ((\boldsymbol{p}_m^t)^\top, \sqrt{2}(\boldsymbol{h}_m^t)^\top)^\top)$. The matrix $\Lambda^\top \Lambda$ is of size $m \times m$, which is independent of the dimension of $\boldsymbol{\mu}$. Therefore, the computational cost to solve problem (10) is negligible since $m$ is usually very small. Here we use the open-source CVXPY library (Diamond & Boyd, 2016) to solve it.

### 3.2 Update Formulations for Gaussian Sampling

Since $\boldsymbol{p}_i^t$ and $\boldsymbol{h}_i^t$ in problem (10) and the update formulation of $\boldsymbol{\mu}$ and $\boldsymbol{\Sigma}$ in Eq. (7) need to calculate expectations of the black-box function. However, those expectations do not have analytical forms, and we estimate them by Monte Carlo sampling.

Specifically, according to Theorem 3.1, the stochastic approximation of $\boldsymbol{p}_i^t$ and $\boldsymbol{h}_i^t$ using Monte Carlo sampling are given as

$$\hat{\boldsymbol{p}}_i^t = \frac{1}{N} \sum\nolimits_{j=1}^N \boldsymbol{\Sigma}_t^{-\frac{1}{2}} (\boldsymbol{x}_j - \boldsymbol{\mu}_t) \big( F_i(\boldsymbol{x}_j) - F_i(\boldsymbol{\mu}_t) \big), \tag{11}$$

$$\hat{\boldsymbol{h}}_i^t = \frac{1}{2N} \sum\nolimits_{j=1}^N \big[ \mathrm{diag}\big( (\boldsymbol{x}_j - \boldsymbol{\mu}_t)(\boldsymbol{x}_j - \boldsymbol{\mu}_t)^\top \boldsymbol{\Sigma}_t^{-1} - \boldsymbol{I} \big) (F_i(\boldsymbol{x}_j) - F_i(\boldsymbol{\mu}_t)) \big], \tag{12}$$

where $\boldsymbol{x}_j$ denotes the $j$-th sample and inspired by Lyu & Tsang (2021), subtracting $F_i(\boldsymbol{\mu}_t)$ is used to improve the computational stability while keeping them as unbiased estimations. By incorporating Theorem 3.1 into Eq. (7), the updated formulations for $\boldsymbol{\mu}$ and $\boldsymbol{\Sigma}$ using Monte Carlo sampling are rewritten as

$$\boldsymbol{\mu}_{t+1} = \boldsymbol{\mu}_t - \beta_t \sum\nolimits_{i=1}^m \lambda_i^t \boldsymbol{\Sigma}_t \hat{\boldsymbol{g}}_i^t, \quad \boldsymbol{\Sigma}_{t+1}^{-1} = \boldsymbol{\Sigma}_t^{-1} + 2\beta_t \sum\nolimits_{i=1}^m \lambda_i^t \hat{G}_i^t, \tag{13}$$

where the stochastic gradients $\hat{\boldsymbol{g}}_i^t$ and $\hat{G}_i^t$ are formulated as

$$\hat{\boldsymbol{g}}_i^t = \frac{1}{N} \sum\nolimits_{j=1}^N \boldsymbol{\Sigma}_t^{-1} (\boldsymbol{x}_j - \boldsymbol{\mu}_t) \big( F_i(\boldsymbol{x}_j) - F_i(\boldsymbol{\mu}_t) \big), \tag{14}$$

$$\hat{G}_i^t = \frac{1}{2N} \sum\nolimits_{j=1}^N \mathrm{diag}\Big[ \boldsymbol{\Sigma}_t^{-1} \big[ \mathrm{diag}\big( (\boldsymbol{x}_j - \boldsymbol{\mu}_t)(\boldsymbol{x}_j - \boldsymbol{\mu}_t)^\top \boldsymbol{\Sigma}_t^{-1} - \boldsymbol{I} \big) (F_i(\boldsymbol{x}_j) - F_i(\boldsymbol{\mu}_t)) \big] \Big]. \tag{15}$$

Note that $\hat{\boldsymbol{g}}_i^t$ is an unbiased estimator for the gradient $\nabla_{\boldsymbol{\mu}} J_i(\boldsymbol{\theta}_t)$ as proved in Lemma B.4. To avoid the scaling problem, in practice, we can employ monotonic transformation for the aggregated objective, more details can be found in Appendix E.

To ensure the convergence, the sequence of weighted vector $\{\boldsymbol{\lambda}^t\}_{t=0}^{T-1}$ is usually required to be a convergent sequence (Zhou et al., 2022b; Fernando et al., 2022; Liu & Vicente, 2021). However, directly solving problem (10) in each iteration cannot guarantee that. Moreover, since solving the composite weights $\boldsymbol{\lambda}^t$ depends on $\hat{\boldsymbol{p}}_i^t$ and $\hat{\boldsymbol{h}}_i^t$, which are related to stochastic gradients $\hat{\boldsymbol{g}}_i^t$ and $\hat{G}_i^t$, the estimation of the composite stochastic gradient may contain some bias, i.e. $\mathbb{E}[\sum_{i=1}^m \lambda_i^t \boldsymbol{\Sigma}_t \hat{\boldsymbol{g}}_i^t] \neq \sum_{i=1}^m \mathbb{E}[\lambda_i^t] \mathbb{E}[\boldsymbol{\Sigma}_t \hat{\boldsymbol{g}}_i^t]$. To generate a stable composite weights sequence and reduce the bias caused by the correlation of weights and the stochastic gradients, we apply a momentum strategy (Zhou et al., 2022b) to $\boldsymbol{\lambda}$. Specifically, in $k$-th iteration, we first solve problem (10) to obtain $\tilde{\boldsymbol{\lambda}}_k$, and then update the weights by $\boldsymbol{\lambda}^k = (1 - \gamma_t)\boldsymbol{\lambda}^{k-1} + \gamma_t \tilde{\boldsymbol{\lambda}}^k$, where $\gamma_t$ is a coefficient and $\gamma_t \in (0, 1]$. To preserve the advantage of maximizing the minimum decrease across all the Gaussian smoothed objectives, the coefficient $\gamma_t$ is set as 1 at the beginning and then decays to 0 as $t \to +\infty$. In Lemma B.5, we show that the bias caused by solving $\boldsymbol{\lambda}^t$ decreases to zero as $\gamma \to 0$.

The complete algorithm is shown in Algorithm 1. Since the computational cost associated with solving problem (10) in each iteration is negligible, the computational cost for the ASMG method per iteration is on the order of $\mathcal{O}(mNd)$.

## 4 Convergence Analysis

In this section, we provide a comprehensive convergence analysis for the proposed ASMG method. All the proofs are put in Appendix C. Firstly, we make a standard assumption for problem (3).

**Assumption 4.1.** The functions $J_i(\boldsymbol{\theta}), \ldots, J_m(\boldsymbol{\theta})$ are $H$-Lipschitz and $L$-smoothness w.r.t. $\boldsymbol{\theta} = \{\boldsymbol{\mu}, \boldsymbol{\Sigma}\} \in \Theta$, where $\Theta := \{\boldsymbol{\mu}, \boldsymbol{\Sigma} \mid \boldsymbol{\mu} \in \mathbb{R}^d, \boldsymbol{\Sigma} \in \mathcal{S}^+\}$.

The smoothness assumption in Assumption 4.1 is widely adopted in MOO (Zhou et al., 2022b; Fernando et al., 2022). Then, we provide a boundedness result for the covariance matrix $\boldsymbol{\Sigma}$.

**Theorem 4.2.** *Suppose that the gradient $\hat{G}_i$ are positive semi-definite matrix, i.e., $\hat{G}_i \succeq \xi \boldsymbol{I}$ for $i = 1, \ldots, m$, where $\xi \geq 0$ and that the covariance matrix is a diagonal matrix. Then for Algorithm 1, we have $\boldsymbol{\Sigma}_T \preceq \frac{\boldsymbol{I}}{\xi \sum_{t=1}^T \beta_t + \boldsymbol{\Sigma}_0^{-1}}$.*

---

**Algorithm 1** The ASMG Method

---

**Require:** number of iterations $T$, step size $\beta$, number of samples $N$.

1: Initialized $\boldsymbol{\theta}_0 = (\boldsymbol{\mu}_0, \boldsymbol{\Sigma}_0)$ and $\gamma_0 = 1$;
2: **for** $t = 0$ to $T - 1$ **do**
3:      Take i.i.d samples $\boldsymbol{z}_j \sim \mathcal{N}(0, I)$ for $j \in \{1, \ldots, N\}$;
4:      Set $\boldsymbol{x}_j = \boldsymbol{\mu}_t + \boldsymbol{\Sigma}_t^{\frac{1}{2}} \boldsymbol{z}_j$ for $j \in \{1, \ldots, N\}$;
5:      Query the batch observations $\{F_1(\boldsymbol{x}_1), \ldots, F_1(\boldsymbol{x}_N), \ldots, F_m(\boldsymbol{x}_1), \ldots, F_m(\boldsymbol{x}_N)\}$;
6:      Query the batch observations $\{F_1(\boldsymbol{\mu}_t), \ldots, F_m(\boldsymbol{\mu}_t)\}$;
7:      Compute $\hat{\boldsymbol{p}}_i^t = \frac{1}{N} \sum_{j=1}^N \boldsymbol{\Sigma}_t^{-\frac{1}{2}}(\boldsymbol{x}_j - \boldsymbol{\mu}_t)(F_i(\boldsymbol{x}_j) - F_i(\boldsymbol{\mu}_t))$;
8:      Compute $\hat{\boldsymbol{h}}_i^t = \frac{1}{2N} \sum_{j=1}^N [\text{diag}((\boldsymbol{x}_j - \boldsymbol{\mu}_t)(\boldsymbol{x}_j - \boldsymbol{\mu}_t)^\top \boldsymbol{\Sigma}_t^{-1} - \boldsymbol{I})(F_i(\boldsymbol{x}_j) - F_i(\boldsymbol{\mu}_t))]$;
9:      Compute $\tilde{\boldsymbol{\lambda}}^t$ by solving the QP problem (10);
10:      Update the weights for the gradient composition $\boldsymbol{\lambda}^t = (1 - \gamma_t)\boldsymbol{\lambda}^{t-1} + \gamma_t \tilde{\boldsymbol{\lambda}}^t$;
11:      Compute the stochastic gradients $\hat{\boldsymbol{g}}_i^t$ and $\hat{\boldsymbol{G}}_i^t$ according to Eqs. (14) (15), respectively;
12:      Set $\boldsymbol{\mu}_{t+1} = \boldsymbol{\mu}_t - \beta_t \sum_{i=1}^m \lambda_i^t \boldsymbol{\Sigma}_t \hat{\boldsymbol{g}}_i^t$ and set $\boldsymbol{\Sigma}_{t+1}^{-1} = \boldsymbol{\Sigma}_t^{-1} + 2\beta_t \sum_{i=1}^m \lambda_i^t \hat{\boldsymbol{G}}_i^t$;
13: **end for**
14: **return** $\boldsymbol{\theta}_T = (\boldsymbol{\mu}_T, \boldsymbol{\Sigma}_T)$.

---

Theorem 4.2 establishes the upper bound for $\boldsymbol{\Sigma}$ throughout the optimization process and is useful to analyze the convergence properties in the non-convex scenario as shown in Section 4.2.

## 4.1 CONVEX CASES

In this section, we assume that each objective in problem (1), i.e., $F_i(\mathbf{x})$ $(i = 1, \ldots, m)$, is convex w.r.t. $\boldsymbol{x}$. Note that the proposed ASMG algorithm approximates the gradients of the objectives of the Gaussian smoothed MOO problem, i.e., problem (3). It is necessary to study the relation between the optimal solutions of the original MOO problem (1) and the corresponding Gaussian-smoothed MOO problem (3), and we put the results in the following proposition.

**Proposition 4.3.** *Suppose $p_{\boldsymbol{\theta}}(\boldsymbol{x})$ is a Gaussian distribution with $\boldsymbol{\theta} = \{\boldsymbol{\mu}, \boldsymbol{\Sigma}\}$ and the functions $F_i(\boldsymbol{x})$, $i = 1, \ldots, m$ are all convex functions. Let $J_i(\boldsymbol{\theta}) = \mathbb{E}_{p_{\boldsymbol{\theta}}}[F_i(\boldsymbol{x})]$. Then for any $\boldsymbol{\lambda} \in \Delta^{m-1}$ and $\boldsymbol{\mu}^* \in \mathcal{X}$, we have $\sum_{i=1}^m \lambda_i(F_i(\boldsymbol{\mu}) - F_i(\boldsymbol{\mu}^*)) \leq \sum_{i=1}^m \lambda_i(J_i(\boldsymbol{\mu}, \boldsymbol{\Sigma}) - J_i(\boldsymbol{\mu}^*, \boldsymbol{0}))$, where $\boldsymbol{0}$ denotes a zero matrix with appropriate size and $J_i(\boldsymbol{\mu}^*, \boldsymbol{0}) = F_i(\boldsymbol{\mu}^*)$.*

When $\boldsymbol{\mu}^*$ is a Pareto-optimal solution of problem (1), Proposition 4.3 implies that the distance to the Pareto-optimal objective values of the original MOO problem is upper-bounded by that of the Gaussian smoothed MOO problem. Then the following theorem captures the convergence of $\boldsymbol{\mu}$ for convex objective functions.

**Theorem 4.4.** *Suppose that $F_i(\boldsymbol{x})$ $(i = 1, \ldots, m)$ is a convex function, $J_i(\boldsymbol{\theta})$ is $c$-strongly convex w.r.t. $\boldsymbol{\mu}$, $\hat{\boldsymbol{G}}_i$ is positive semi-definite matrix such that $\xi\boldsymbol{I} \preceq \hat{\boldsymbol{G}}_i \preceq \frac{c\boldsymbol{I}}{4}$ with $\xi \geq 0$, $\boldsymbol{\Sigma}_0 \in \mathcal{S}^+$, and $\boldsymbol{\Sigma}_0 \preceq R\boldsymbol{I}$ where $R > 0$. If $\beta \leq \frac{1}{L}$ and the sequence $\{\boldsymbol{\mu}_t\}$ generated by Algorithm 1 satisfies that the distance between the sequence $\{\boldsymbol{\mu}_t\}$ and the Pareto set is bounded, i.e., $\|\boldsymbol{\mu}_t - \boldsymbol{\mu}^*\| \leq D$ where $\boldsymbol{\mu}^*$ denotes a Pareto optimal solution of problem (1), then with Assumption 4.1, we have*

$$\frac{1}{T}\sum_{t=0}^{T-1}\mathbb{E}_{\boldsymbol{z}}\left[\sum_{i=1}^m \lambda_i^t(J_i(\boldsymbol{\mu}_{t+1}, \boldsymbol{\Sigma}_t) - J_i(\boldsymbol{\mu}^*, 0))\right] = \mathcal{O}\left(\frac{1}{\beta T} + \frac{\log T}{T} + \gamma\right). \quad (16)$$

Based on Theorem 4.4 and Proposition 4.3, when $\beta = \mathcal{O}(1)$ and $\gamma = \mathcal{O}(T^{-1})$, we have

$$\frac{1}{T}\sum_{t=0}^{T-1}\mathbb{E}_{\boldsymbol{z}}\left[\sum_{i=1}^m \lambda_i^t(F_i(\boldsymbol{\mu}_{t+1}) - F_i(\boldsymbol{\mu}^*))\right] = \mathcal{O}(\frac{\log T}{T}). \quad (17)$$

Therefore, the proposed ASMG algorithm possesses a convergence rate $\mathcal{O}(\frac{\log T}{T})$ in convex cases. Note that Theorem 4.4 does not require each objective function $F_i(\boldsymbol{x})$ to be differentiable. Hence, Theorem 4.4 holds for non-smooth convex functions $\{F_i(\boldsymbol{x})\}$. If $F_i(\boldsymbol{x})$ is $c$-strongly convex, then $J_i(\boldsymbol{\theta})$ is also $c$-strongly convex (Domke, 2020) and Theorem 4.4 holds.

## 4.2 NON-CONVEX CASES

In many practical problems, the objective functions of problem (1) are non-convex, and we aim to find a Pareto stationary solution. Similar to Proposition 4.3, we have the following result to reveal the relation between Pareto stationary solutions of problems (1) and (3).

**Proposition 4.5.** *Suppose $p_{\boldsymbol{\theta}}(\boldsymbol{x})$ is a Gaussian distribution with $\boldsymbol{\theta} = \{\boldsymbol{\mu}, \boldsymbol{\Sigma}\}$ and $F_i(\boldsymbol{x})$ ($i = 1, \ldots, m$) is a $L_F$-Lipschitz smooth function. Let $J_i(\boldsymbol{\theta}) = \mathbb{E}_{p_{\boldsymbol{\theta}}}[F_i(\boldsymbol{x})]$ and $\boldsymbol{\Sigma}$ be a diagonal matrix. If $\boldsymbol{\mu}^*$ is a Pareto stationary solution of problem (3) and there exists $\boldsymbol{\lambda} \in \Delta^{m-1}$ such that $\|\sum_{i=1}^{m} \lambda_i \nabla_{\boldsymbol{\mu}} J_i(\boldsymbol{\mu}^*)\| = 0$, then we have $\|\sum_{i=1}^{m} \lambda_i \nabla F_i(\boldsymbol{\mu}^*)\|^2 \leq L_F^2 \|\mathrm{diag}(\boldsymbol{\Sigma})\|_1$ and this implies that $\boldsymbol{\mu}^*$ is a $\epsilon$-accurate Pareto stationary solution of problem (1) with $\epsilon = L_F^2 \|\mathrm{diag}(\boldsymbol{\Sigma})\|_1$.*

According to Proposition 4.5, a Pareto stationary solution of problem (3) is a $\epsilon$-accurate Pareto stationary solution of problem (1). The following theorem establishes the convergence of the proposed ASMG method under the non-convex case.

**Theorem 4.6.** *Suppose that $J_i(\boldsymbol{\theta})$ ($i = 1, \ldots, m$) is bounded, i.e., $|J_i(\boldsymbol{\theta})| \leq B$, $\hat{G}_i$ is positive semi-definite matrix such that $\xi \boldsymbol{I} \preceq \hat{G}_i \preceq b\boldsymbol{I}$ with $b \geq \xi > 0$, $\boldsymbol{\Sigma}_0 \in \mathcal{S}^+$, and $\boldsymbol{\Sigma}_0 \preceq R\boldsymbol{I}$ with $R > 0$. If $\beta \leq \frac{1}{RL\sqrt{d}}$, then with Assumption 4.1 we have*

$$\frac{1}{T} \sum\nolimits_{t=0}^{T-1} \mathbb{E}_{\boldsymbol{z}} \left[ \left\| \sum\nolimits_{i=1}^{m} \lambda_i^t \nabla_{\boldsymbol{\mu}} J_i(\boldsymbol{\theta}_t) \right\|^2 \right] = \mathcal{O}\left( \frac{\gamma}{\beta} + \frac{1}{\beta T} + \gamma + \beta \right). \tag{18}$$

According to Theorem 4.6, if $\beta = \mathcal{O}(T^{-\frac{1}{2}})$ and $\gamma = \mathcal{O}(T^{-1})$, the proposed ASMG method possesses a $\mathcal{O}(T^{-\frac{1}{2}})$ convergence rate to reach a Pareto stationary solution for problem (3), which is a $\epsilon$-accurate Pareto stationary solution of problem (1). According to Theorem 4.2, when $\beta = \mathcal{O}(T^{-\frac{1}{2}})$, diagonal entries of $\boldsymbol{\Sigma}_T$ converge to zero as $T \to \infty$ and hence $\epsilon \to 0$, leading to a Pareto stationary solution for problem (1).

## 5 RELATED WORKS

Several kinds of approaches have been studied for black-box optimization, such as Bayesian optimization (BO) (Srinivas et al., 2009; Lyu et al., 2019), evolution strategies (ES) (Back, 1991; Hansen, 2006), and genetic algorithms (GA) (Srinivas & Patnaik, 1994). BO-based methods are inefficient in handling high-dimensional problems and GA methods lack convergence analysis. ES-based methods are better for relatively high-dimensional problems and have been applied in many applications such as reinforcement learning (Liu et al., 2019) and prompt learning (Sun et al., 2022b;a). Although BO achieves good query efficiency for low-dimensional problems, it often fails to handle high-dimensional problems with large sample budgets (Eriksson et al., 2019). The computation of GP with a large number of samples itself is expensive, and the internal optimization of the acquisition functions is challenging.

Among ES-based methods, CMA-ES (Hansen, 2006) is a representative one. It uses second-order information to search candidate solutions by updating the mean and covariance matrix of the likelihood of candidate distributions. The CMA-ES method is widely adopted in many learning tasks (Won et al., 2017; Sun et al., 2022b;a; Han et al., 2023). Though it is designed for single-objective black-box optimization, it is also applied to black-box multi-task learning (Sun et al., 2023), where all objectives are aggregated with equal weights. Therefore, we consider CMA-ES as an important baseline method in our experiments.

## 6 EMPIRICAL STUDY

In this section, we empirically evaluate the proposed ASMG method on different problems. The experiments are conducted on a single NVIDIA GeForce RTX 3090 GPU.

### 6.1 SYNTHETIC PROBLEMS

We compare the proposed ASMG method with CMA-ES (Hansen, 2006), ES (Salimans et al., 2017), BES (Gao & Sener, 2022), and MMES (He et al., 2020) methods on the following three $d$-dimensional

synthetic benchmark test problems:

$$F(\boldsymbol{x}) = \Big( \sum_{i=1}^{d} 10^{\frac{2(i-1)}{d-1}} |\boldsymbol{x}_i - 0.01|, \sum_{i=1}^{d} 10^{\frac{2(i-1)}{d-1}} |\boldsymbol{x}_i + 0.01| \Big), \tag{19}$$

$$F(\boldsymbol{x}) = \Big( \sum_{i=1}^{d} |\boldsymbol{x}_i - 0.1|^{\frac{1}{2}}, \sum_{i=1}^{d} |\boldsymbol{x}_i + 0.1|^{\frac{1}{2}} \Big), \tag{20}$$

$$F(\boldsymbol{x}) = \Big( \sum_{i=1}^{d} 10^{\frac{2(i-1)}{d-1}} |\boldsymbol{x}_i|^{\frac{1}{2}}, 10d + \sum_{i=1}^{d} \big( (10^{\frac{(i-1)}{d-1}} \boldsymbol{x}_i)^2 - 10 \cos(2\pi 10^{\frac{(i-1)}{d-1}} \boldsymbol{x}_i) \big) \Big). \tag{21}$$

Test problems (19)-(21) are called the shift $l_1$-ellipsoid, shift $l_{\frac{1}{2}}$-ellipsoid, and mixed ellipsoid-rastrigin 10, respectively.

For the baseline methods, by following Sun et al. (2023), we aggregate multiple objectives with equal weights to become a single objective. The results are evaluated by calculating the Euclidean distance between the solution $\boldsymbol{x}$ and the optimal solution set $\mathcal{P}$, i.e., $\mathcal{E} = \text{dist}(\boldsymbol{x}, \mathcal{P})$. *Due to the page limitation, the details of the evaluation metric and implementation are put in Appendix F.1.*

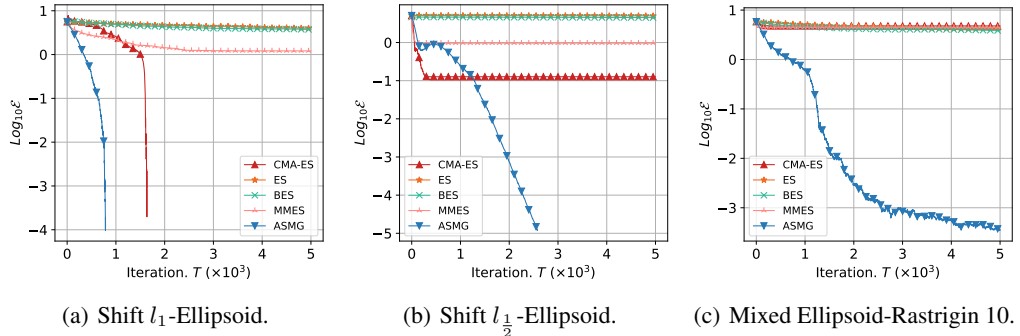

(a) Shift $l_1$-Ellipsoid.     (b) Shift $l_{\frac{1}{2}}$-Ellipsoid.     (c) Mixed Ellipsoid-Rastrigin 10.

Figure 1: Results on the synthetic problems with 50 samples (i.e., $N = 50$).

**Results.** Figure 1 shows the results on those three $d$-dimensional synthetic problems with 50 samples (i.e., $N = 50$) and $d = 100$. The proposed ASMG method approximately achieves a linear convergence rate in the logarithm scale and can arrive at solutions with a high precision, i.e., $10^{-4}$, on three cases. The CMA-ES method can converge with high precision on problem (19) but only achieve $10^{-1}$ precision on problem (20). The MMES method also cannot reach a high precision on these problems. Moreover, the CMA-ES and MMES methods fail on problem (21) and the ES and BES methods fail on all three problems. The results show that it could be challenging for ES and BES to optimize non-smooth or non-convex test functions without adaptively updating mean and covariance. Those results consistently demonstrate the effectiveness of the proposed ASMG method.

## 6.2 BLACK-BOX MULTI-TASK LEARNING

In this section, we apply the proposed ASMG method to black-box multi-task learning. Multi-task learning (MTL) (Caruana, 1997; Zhang & Yang, 2022) is a widely adopted paradigm and aims to train a single model to handle multiple tasks simultaneously. Given $m$ tasks, task $i$ has a training dataset $\mathcal{D}_i$. Let $\mathcal{L}_i(\mathcal{D}_i; \mathcal{M}_\Phi)$ denote the average loss on $\mathcal{D}_i$ for task $i$ using the model $\mathcal{M}$ with parameter $\Phi$. Then MTL can be formulated as a MOO problem with $m$ objectives as $\min_\Phi(\mathcal{L}_i, \dots, \mathcal{L}_m)$. For the conventional MTL setting, the model $\mathcal{M}$ is available for backward propagation, allowing the optimization problem to be solved using the gradients of the model parameters, i.e., $\nabla_\Phi \mathcal{L}_i$. However, in many practical scenarios, such as multi-task prompt tuning for extremely large pre-trained models (Sun et al., 2023), part of the model $\mathcal{M}$ remains fixed in the service and is only accessible through an inference API. This results in the gradient of the objectives $\mathcal{L}_i$ with respect to the local parameters $\phi \subset \Phi$ being unavailable. For cases where the gradients of task losses with respect to the learned parameter $\phi$ cannot be explicitly calculated in MTL, we refer to black-box MTL.

**Problem Formulation.** We consider a specific black-box MTL problem, where our focus is to learn a shared prompt for all tasks using pre-trained vision-language models (Sun et al., 2023; Wang et al., 2023; Liu et al., 2023). Following the setup in Liu et al. (2023), we employ the CLIP model (Radford et al., 2021) as the base model. In this context, our model $\mathcal{M}$ can be expressed as $\mathcal{M} = \{\mathcal{M}_c, \boldsymbol{p}\}$, where $\mathcal{M}_c$ represents the fixed CLIP model, and $\boldsymbol{p} \in \mathbb{R}^D$ is the token embedding of the prompt. Note

that the CLIP model is treated as a black-box model, making it impossible to calculate the gradient of the token embedding $p$ in the text encoder using backward propagation. Inspired by Sun et al. (2022b;a), we optimize $v \in \mathbb{R}^d$ and employ a fixed randomly initialized matrix $A \in \mathbb{R}^{D \times d}$ to project $v$ onto the token embedding space instead of directly optimizing the prompt $p$. Consequently, the corresponding black-box MTL problem can be formulated as

$$\min_{v \in \mathbb{R}^d} \left( \mathcal{L}_1(\mathcal{D}_1; \{\mathcal{M}_c, Av\}), \ldots, \mathcal{L}_m(\mathcal{D}_m; \{\mathcal{M}_c, Av\}) \right). \tag{22}$$

*The details of the CLIP model with token embedding and the loss function $\mathcal{L}_i$ are put in Appendix F.2.*

**Baselines.** The proposed ASMG method is compared with (i) the zero-shot setting, which evaluates the model on the downstream datasets that were not seen during the training phase without prompt tuning (Radford et al., 2021);   (ii) four ES-based black-box optimization methods, i.e., ES (Salimans et al., 2017), BES (Gao & Sener, 2022), MMES (He et al., 2020) and CMA-ES (Hansen, 2006), where we simply transform multiple objectives into one single objective by equal weights;   (iii) the ASMG-EW method, where we fix the weighted vector as $\lambda^t = \frac{1}{m}$ for ASMG during optimization. *The implementation details are put in Appendix F.2.*

Table 1: Results on the *Office-31* and *Office-home* datasets. Each experiment is repeated over 3 random seeds and the mean classification accuracy (%) is reported. The best result across all groups is in **bold** and the best result in each comparison group is underlined.

| Method | Office-31 | | | | Office-home | | | | |
|---|---|---|---|---|---|---|---|---|---|
| | A | D | W | Avg | Ar | Cl | Pr | Rw | Avg |
| Zero-shot | 73.68 | 79.51 | 66.67 | 73.28 | 73.06 | 51.68 | 83.79 | 81.41 | 72.48 |
| | *Dimension $d = 256$* | | | | | | | | |
| ES | 75.10 | 80.05 | 71.67 | $75.61_{\pm 1.18}$ | 71.16 | 47.96 | 80.33 | 80.90 | $70.09_{\pm 0.51}$ |
| BES | 72.65 | 80.60 | 73.52 | $75.59_{\pm 0.90}$ | 68.94 | 45.97 | 81.53 | 79.42 | $68.97_{\pm 1.40}$ |
| MMES | 75.90 | 83.33 | 76.67 | $78.63_{\pm 0.59}$ | 71.85 | 49.26 | 82.10 | 81.37 | $71.14_{\pm 0.69}$ |
| CMA-ES | 76.24 | 87.98 | 75.93 | $80.05_{\pm 1.34}$ | 69.26 | 50.09 | 85.73 | 82.13 | $71.80_{\pm 0.22}$ |
| ASMG-EW | 76.52 | 83.88 | 77.22 | $79.21_{\pm 1.20}$ | 70.02 | 47.13 | 80.23 | 79.50 | $69.22_{\pm 1.39}$ |
| ASMG | 77.83 | 86.61 | 80.56 | $\mathbf{81.67}_{\pm 0.64}$ | 74.26 | 53.52 | 86.23 | 83.03 | $\mathbf{74.26}_{\pm 1.06}$ |
| | *Dimension $d = 512$* | | | | | | | | |
| ES | 75.95 | 81.69 | 75.37 | $77.67_{\pm 0.91}$ | 70.78 | 48.39 | 82.06 | 81.15 | $70.60_{\pm 0.36}$ |
| BES | 75.73 | 82.51 | 74.81 | $77.68_{\pm 1.88}$ | 69.39 | 48.18 | 82.94 | 80.29 | $70.20_{\pm 0.51}$ |
| MMES | 76.01 | 84.70 | 77.22 | $79.31_{\pm 0.34}$ | 70.40 | 50.45 | 85.10 | 82.56 | $72.13_{\pm 1.19}$ |
| CMA-ES | 76.75 | 87.16 | 77.22 | $80.38_{\pm 0.48}$ | 70.46 | 50.02 | 86.26 | 82.02 | $72.19_{\pm 0.27}$ |
| ASMG-EW | 78.01 | 84.70 | 76.67 | $79.79_{\pm 1.45}$ | 69.20 | 46.91 | 80.51 | 80.68 | $69.33_{\pm 0.52}$ |
| ASMG | 78.63 | 87.43 | 78.33 | $81.47_{\pm 0.37}$ | 73.50 | 52.84 | 85.88 | 83.78 | $74.00_{\pm 0.81}$ |
| | *Dimension $d = 1024$* | | | | | | | | |
| ES | 72.59 | 78.14 | 74.81 | $75.18_{\pm 1.91}$ | 70.34 | 47.38 | 82.59 | 80.54 | $70.21_{\pm 0.08}$ |
| BES | 72.14 | 79.51 | 71.67 | $74.44_{\pm 0.84}$ | 70.27 | 48.25 | 79.94 | 80.00 | $69.62_{\pm 0.81}$ |
| MMES | 77.09 | 81.42 | 75.74 | $78.09_{\pm 0.95}$ | 71.03 | 49.19 | 84.29 | 81.95 | $71.61_{\pm 0.41}$ |
| CMA-ES | 76.87 | 87.16 | 77.59 | $80.54_{\pm 0.41}$ | 71.28 | 50.92 | 85.73 | 82.49 | $72.61_{\pm 0.39}$ |
| ASMG-EW | 77.15 | 82.51 | 77.78 | $79.15_{\pm 1.48}$ | 69.20 | 47.09 | 81.36 | 80.86 | $69.63_{\pm 0.88}$ |
| ASMG | 76.30 | 87.70 | 80.19 | $81.40_{\pm 0.49}$ | 73.18 | 51.82 | 85.84 | 83.21 | $73.51_{\pm 0.07}$ |

**Results.** Table 1 presents experimental results on the *Office-31* and *Office-home* datasets for three different dimensions of $z$. We can see that the ASMG method consistently outperforms all baselines in terms of average classification accuracy across different settings, highlighting its effectiveness. When comparing ASMG with ASMG-EW, the results demonstrate the effectiveness of adaptive stochastic gradient. Notably, even in the high-dimensional setting (i.e., $d = 1024$), our method maintains good performance. Remarkably, ASMG achieves the highest average classification accuracy when $d = 256$, surpassing zero-shot by 8.4% on *Office-31* and 1.8% on *Office-home*. This further validates the effectiveness of the proposed ASMG method.

## 7 CONCLUSION

In this paper, we propose ASMG, a novel and effective adaptive stochastic gradient-based method for solving the black-box MOO problem. Specifically, we smooth the black-box MOO problem to a Gaussian smoothed MOO and we propose a novel adaptive stochastic gradient approximation approach to solve it. Theoretically, we explore the connections between the MOO and the corresponding Gaussian smoothed MOO, and we provide a convergence guarantee for ASMG under both convex and non-convex scenarios. Moreover, empirical studies on synthetic problems and black-box MTL demonstrate the effectiveness of the proposed ASMG method.

## ACKNOWLEDGEMENTS

This work is supported by National Key R&D Program of China 2022ZD0160300, NSFC key grant under grant no. 62136005, NSFC general grant under grant no. 62076118, and Shenzhen fundamental research program JCYJ20210324105000003.

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

# APPENDIX

## A  PROOF OF THE RESULT IN SECTION 3.1

### A.1  PROOF OF UPDATED RULE

The objective of the inner minimization of problem (6) can be rewritten as

$$\left\langle \sum_{i=1}^{m} \lambda_i \nabla_{\boldsymbol{\theta}} J_i(\boldsymbol{\theta}_t), \boldsymbol{\theta} \right\rangle + \frac{1}{\beta_t} \mathrm{KL}(p_{\boldsymbol{\theta}} \| p_{\boldsymbol{\theta}_t}) = \boldsymbol{\mu}^{\top} (\sum_{i=1}^{m} \lambda_i \nabla_{\boldsymbol{\mu}} J_i(\boldsymbol{\theta}_t)) + \sum_{i=1}^{m} \lambda_i \mathrm{tr}(\boldsymbol{\Sigma} \nabla_{\boldsymbol{\Sigma}} J_i(\boldsymbol{\theta}_t))$$

$$+ \frac{1}{2\beta_t} \left[ \mathrm{tr}(\boldsymbol{\Sigma}_t^{-1} \boldsymbol{\Sigma}) + (\boldsymbol{\mu} - \boldsymbol{\mu}_t)^{\top} \boldsymbol{\Sigma}_t^{-1} (\boldsymbol{\mu} - \boldsymbol{\mu}_t) + \log \frac{|\boldsymbol{\Sigma}_t|}{|\boldsymbol{\Sigma}|} - d \right], \quad (23)$$

where $\nabla_{\boldsymbol{\mu}} J_i(\boldsymbol{\theta}_t)$ and $\nabla_{\boldsymbol{\Sigma}} J_i(\boldsymbol{\theta}_t)$ denotes the derivative w.r.t $\boldsymbol{\mu}$ and $\boldsymbol{\Sigma}$ taking at $\boldsymbol{\mu} = \boldsymbol{\mu}_t$ and $\boldsymbol{\Sigma} = \boldsymbol{\Sigma}_t$, respectively. We can see the above problem is convex with respect to $\boldsymbol{\mu}$ and $\boldsymbol{\Sigma}$. Taking the derivative w.r.t $\boldsymbol{\mu}$ and $\boldsymbol{\Sigma}$ and setting them to zero, we can obtain that

$$\sum_{i=1}^{m} \lambda_i \nabla_{\boldsymbol{\mu}} J_i(\boldsymbol{\theta}_t) + \frac{1}{\beta_t} \boldsymbol{\Sigma}_t^{-1} (\boldsymbol{\mu} - \boldsymbol{\mu}_t) = 0, \quad (24)$$

$$\sum_{i=1}^{m} \lambda_i \nabla_{\boldsymbol{\Sigma}} J_i(\boldsymbol{\theta}_t) + \frac{1}{2\beta_t} [\boldsymbol{\Sigma}_t^{-1} - \boldsymbol{\Sigma}^{-1}] = 0. \quad (25)$$

Substituting the above equalities into the regularization term of the objective of the outer optimization problem we have

$$\frac{1}{\beta_t} \mathrm{KL}(p_{\boldsymbol{\theta}} \| p_{\boldsymbol{\theta}_t}) = \frac{1}{2\beta_t} \left[ \mathrm{tr}(\boldsymbol{\Sigma}_t^{-1} \boldsymbol{\Sigma}) + (\boldsymbol{\mu} - \boldsymbol{\mu}_t)^{\top} \boldsymbol{\Sigma}_t^{-1} (\boldsymbol{\mu} - \boldsymbol{\mu}_t) + \log \frac{|\boldsymbol{\Sigma}_t|}{|\boldsymbol{\Sigma}|} - d \right] \quad (26)$$

$$= \frac{1}{2\beta_t} \mathrm{tr}\left( \boldsymbol{I} - 2\beta_t \sum_{i=1}^{m} \lambda_i \nabla_{\boldsymbol{\Sigma}} J_i(\boldsymbol{\theta}_t) \boldsymbol{\Sigma} \right) - \frac{1}{2} (\boldsymbol{\mu} - \boldsymbol{\mu}_t)^{\top} \sum_{i=1}^{m} \lambda_i \nabla_{\boldsymbol{\mu}} J_i(\boldsymbol{\theta}_t)$$

$$+ \frac{1}{2\beta_t} \log \left( |\boldsymbol{\Sigma}_t (\boldsymbol{\Sigma}_t^{-1} + 2\beta_t \sum_{i=1}^{m} \lambda_i \nabla_{\boldsymbol{\Sigma}} J_i(\boldsymbol{\theta}_t))| \right) - \frac{d}{2\beta_t} \quad (27)$$

$$= \frac{d}{2\beta_t} - \left\langle \sum_{i=1}^{m} \lambda_i \nabla_{\boldsymbol{\Sigma}} J_i(\boldsymbol{\theta}_t), \boldsymbol{\Sigma} \right\rangle - \frac{1}{2} (\boldsymbol{\mu} - \boldsymbol{\mu}_t)^{\top} \sum_{i=1}^{m} \lambda_i \nabla_{\boldsymbol{\mu}} J_i(\boldsymbol{\theta})$$

$$+ \frac{1}{2\beta_t} \log(|\boldsymbol{I} + 2\beta_t \boldsymbol{\Sigma}_t \sum_{i=1}^{m} \lambda_i \nabla_{\boldsymbol{\Sigma}} J_i(\boldsymbol{\theta}_t)|) - \frac{d}{2\beta_t} \quad (28)$$

$$= -\left\langle \sum_{i=1}^{m} \lambda_i \nabla_{\boldsymbol{\Sigma}} J_i(\boldsymbol{\theta}_t), \boldsymbol{\Sigma} - \boldsymbol{\Sigma}_t \right\rangle - \frac{1}{2} (\boldsymbol{\mu} - \boldsymbol{\mu}_t)^{\top} \sum_{i=1}^{m} \lambda_i \nabla_{\boldsymbol{\mu}} J_i(\boldsymbol{\theta}) + \frac{Q_t}{2\beta_t} \quad (29)$$

where $Q_t$ is given as below.

$$Q_t = \log(|\boldsymbol{I} + 2\beta_t \boldsymbol{\Sigma}_t \sum_{i=1}^{m} \lambda_i \nabla_{\boldsymbol{\Sigma}} J_i(\boldsymbol{\theta}_t)|) - 2\beta_t \left\langle \sum_{i=1}^{m} \lambda_i \nabla_{\boldsymbol{\Sigma}} J_i(\boldsymbol{\theta}_t), \boldsymbol{\Sigma}_t \right\rangle \quad (30)$$

$$= \log(|\boldsymbol{I} + 2\beta_t \boldsymbol{\Sigma}_t \sum_{i=1}^{m} \lambda_i \nabla_{\boldsymbol{\Sigma}} J_i(\boldsymbol{\theta}_t)|) - \mathrm{tr}(2\beta_t \boldsymbol{\Sigma}_t \sum_{i=1}^{m} \lambda_i \nabla_{\boldsymbol{\Sigma}} J_i(\boldsymbol{\theta}_t)). \quad (31)$$

Since $\boldsymbol{\Sigma}_t$ and $\nabla_{\boldsymbol{\Sigma}} J_i(\boldsymbol{\theta}_t)$ are both diagonal matrix, we denote $\mathrm{diag}(\boldsymbol{\Sigma}_t \sum_{i=1}^{m} \lambda_i \nabla_{\boldsymbol{\Sigma}} J_i(\boldsymbol{\theta}_t)) = (v_t^1, \ldots, v_t^d)$ in $t$-th iteration, then we have

$$Q_t = \log(\prod_{i=1}^{d} (1 + 2\beta_t v_t^i)) - \sum_{i=1}^{d} 2\beta_t v_t^i = \sum_{i=1}^{d} \left( \log(1 + 2\beta_t v_t^i) - 2\beta_t v_t^i \right) = \sum_{i=1}^{d} -2\beta_t^2 (v_t^i)^2 + \mathcal{O}(\beta_t^3 (v_t^i)^3),$$

$$(32)$$

where the last equality is due to the Taylor expansion. Note that $\mathcal{O}(\beta_t^3 (v_t^i)^3)$ decrease to zero when $\beta_t \to 0$. We can approximate $Q_t$ by $\sum_{i=1}^d -2\beta_t^2 (v_t^i)^2$. Then substituting Eqs. (24) (29) and $Q_t$ into the outer optimization problem of problem (6), we have

$$\left\langle \sum_{i=1}^m \lambda_i \nabla_{\boldsymbol{\theta}} J_i(\boldsymbol{\theta}_t), \boldsymbol{\theta} - \boldsymbol{\theta}_t \right\rangle + \frac{1}{\beta_t} \mathrm{KL}(p_{\boldsymbol{\theta}} \| p_{\boldsymbol{\theta}_t}) \tag{33}$$

$$= \left\langle \sum_{i=1}^m \lambda_i \nabla_{\boldsymbol{\mu}} J_i(\boldsymbol{\theta}_t), \boldsymbol{\mu} - \boldsymbol{\mu}_t \right\rangle + \left\langle \sum_{i=1}^m \lambda_i \nabla_{\boldsymbol{\Sigma}} J_i(\boldsymbol{\theta}_t), \boldsymbol{\Sigma} - \boldsymbol{\Sigma}_t \right\rangle + \frac{1}{\beta_t} \mathrm{KL}(p_{\boldsymbol{\theta}} \| p_{\boldsymbol{\theta}_t}) \tag{34}$$

$$= \frac{1}{2} \left\langle \sum_{i=1}^m \lambda_i \nabla_{\boldsymbol{\mu}} J_i(\boldsymbol{\theta}_t), \boldsymbol{\mu} - \boldsymbol{\mu}_t \right\rangle + \frac{Q_t}{2\beta_t} \tag{35}$$

$$= -\frac{\beta_t}{2} \left\langle \sum_{i=1}^m \lambda_i \nabla_{\boldsymbol{\mu}} J_i(\boldsymbol{\theta}_t), \boldsymbol{\Sigma}_t \sum_{i=1}^m \lambda_i \nabla_{\boldsymbol{\mu}} J_i(\boldsymbol{\theta}_t) \right\rangle - \beta_t \left\langle \boldsymbol{\Sigma}_t \sum_{i=1}^m \lambda_i \nabla_{\boldsymbol{\Sigma}} J_i(\boldsymbol{\theta}_t), \boldsymbol{\Sigma}_t \sum_{i=1}^m \lambda_i \nabla_{\boldsymbol{\Sigma}} J_i(\boldsymbol{\theta}_t) \right\rangle. \tag{36}$$

Therefore, the outer optimization problem is equivalent to the following problem

$$\min_{\boldsymbol{\lambda}^t \in \Delta^{m-1}} \left\| \boldsymbol{\Sigma}_t^{\frac{1}{2}} \sum_{i=1}^m \lambda_i \nabla_{\boldsymbol{\theta}} J_i(\boldsymbol{\theta}_t) \right\|^2 + 2 \left\| \mathrm{diag}(\boldsymbol{\Sigma}_t \sum_{i=1}^m \lambda_i \nabla_{\boldsymbol{\Sigma}} J_i(\boldsymbol{\theta}_t)) \right\|^2, \tag{37}$$

where we reach the result in Eq. (10).

### A.2 PROOF OF THEOREM 3.1

We now provide the proof of the gradient of $\mathbb{E}_{p_{\boldsymbol{\theta}}}[F_i(\boldsymbol{x})]$ w.r.t $\boldsymbol{\mu}$ and $\boldsymbol{\Sigma}$.

$$\nabla_{\boldsymbol{\mu}} \mathbb{E}_{p_{\boldsymbol{\theta}}}[F_i(\boldsymbol{x})] = \mathbb{E}_{p_{\boldsymbol{\theta}}}[F_i(\boldsymbol{x}) \nabla_{\boldsymbol{\mu}} \log(p(\boldsymbol{x}; \boldsymbol{\mu}, \boldsymbol{\Sigma}))] \tag{38}$$

$$= \mathbb{E}_{p_{\boldsymbol{\theta}}}[F_i(\boldsymbol{x}) \nabla_{\boldsymbol{\mu}}(-\frac{1}{2}(\boldsymbol{x} - \boldsymbol{\mu})^\top \boldsymbol{\Sigma}^{-1} (\boldsymbol{x} - \boldsymbol{\mu})] \tag{39}$$

$$= \mathbb{E}_{p_{\boldsymbol{\theta}}}[\boldsymbol{\Sigma}^{-1}(\boldsymbol{x} - \boldsymbol{\mu}) F_i(\boldsymbol{x})]. \tag{40}$$

We further have

$$\nabla_{\boldsymbol{\Sigma}} \mathbb{E}_{p_{\boldsymbol{\theta}}}[F_i(\boldsymbol{x})] = \mathbb{E}_{p_{\boldsymbol{\theta}}}[F_i(\boldsymbol{x}) \nabla_{\boldsymbol{\Sigma}} \log(p(\boldsymbol{x}; \boldsymbol{\mu}, \boldsymbol{\Sigma}))] \tag{41}$$

$$= \mathbb{E}_{p_{\boldsymbol{\theta}}}[F_i(\boldsymbol{x}) \nabla_{\boldsymbol{\Sigma}}(-\frac{1}{2}(\boldsymbol{x} - \boldsymbol{\mu})^\top \boldsymbol{\Sigma}^{-1} (\boldsymbol{x} - \boldsymbol{\mu}) - \frac{1}{2} \log \det(\boldsymbol{\Sigma}))] \tag{42}$$

$$= \frac{1}{2} \mathbb{E}_{p_{\boldsymbol{\theta}}}[(\boldsymbol{\Sigma}^{-1}(\boldsymbol{x} - \boldsymbol{\mu})(\boldsymbol{x} - \boldsymbol{\mu})^\top \boldsymbol{\Sigma}^{-1} - \boldsymbol{\Sigma}^{-1}) F_i(\boldsymbol{x})], \tag{43}$$

where we reach the conclusion.

## B TECHNICAL LEMMAS

In this section, we introduce the following technical lemmas for analysis. The proof of all technical lemmas is put in Appendix D.

**Lemma B.1.** *Suppose $\boldsymbol{\Sigma}$ and $\hat{\boldsymbol{\Sigma}}$ are two $d$-dimensional diagonal matrix and $\boldsymbol{z}$ is a $d$-dimensional vector, then we have $\|\boldsymbol{\Sigma} \boldsymbol{z}\| \leq \|\boldsymbol{\Sigma}\|_F \|\boldsymbol{z}\|$ and $\|\boldsymbol{\Sigma} \hat{\boldsymbol{\Sigma}}\|_F \leq \|\boldsymbol{\Sigma}\|_F \|\hat{\boldsymbol{\Sigma}}\|_F$.*

**Lemma B.2.** *Given a convex function $f(\boldsymbol{x})$, for Gaussian distribution with parameters $\boldsymbol{\theta} := \{\boldsymbol{\mu}, \boldsymbol{\Sigma}^{\frac{1}{2}}\}$, let $\bar{J}(\boldsymbol{\theta}) := \mathbb{E}_{p(\boldsymbol{x}; \boldsymbol{\theta})}[f(\boldsymbol{x})]$. Then $\bar{J}(\boldsymbol{\theta})$ is a convex function with respect to $\boldsymbol{\theta}$.*

**Lemma B.3.** *Suppose that the gradient $\hat{G}_i$ are positive semi-definite matrix and satisfies $\xi \boldsymbol{I} \preceq \hat{G}_i \preceq b\boldsymbol{I}$. Then for algorithm 1, we have the following results.*

    *(a) The (diagonal) covariance matrix $\boldsymbol{\Sigma}_T$ satisfies*

$$\frac{1}{2b \sum_{t=1}^T \beta_t \boldsymbol{I} + \boldsymbol{\Sigma}_0^{-1}} \preceq \boldsymbol{\Sigma}_T \preceq \frac{1}{2\xi \sum_{t=1}^T \beta_t \boldsymbol{I} + \boldsymbol{\Sigma}_0^{-1}}.$$

(b) $\|\mathbf{\Sigma}_t\|_F \leq \frac{\sqrt{d}}{2\xi \sum_{t=1}^T \beta_t}$.

(c) $\|\mathbf{\Sigma}_{t+1} - \mathbf{\Sigma}_t\|_F \leq \frac{b\beta_t d^{\frac{3}{2}}}{2\xi^2(\sum_{t=1}^T \beta_t)^2}$.

**Lemma B.4.** *Suppose the gradient estimator $\hat{\boldsymbol{g}}_i^t$ for the $i$-th objective in $t$-th iteration as*

$$\hat{\boldsymbol{g}}_i^t = \mathbf{\Sigma}_t^{-\frac{1}{2}} \boldsymbol{z} \big( F_i(\boldsymbol{\mu}_t + \mathbf{\Sigma}_t^{\frac{1}{2}} \boldsymbol{z}) - F_i(\boldsymbol{\mu}_t) \big),$$

*where $\boldsymbol{z} \sim \mathcal{N}(0, I)$. Suppose assumption 4.1 holds, the gradient $\hat{G}_i$ are positive semi-definite matrix and satisfies $\xi \boldsymbol{I} \preceq \hat{G}_i \preceq b\boldsymbol{I}$ and $\mathbf{\Sigma}_0 \preceq R\boldsymbol{I}$, where $\xi, b, R \geq 0$. Then we have*

(a) *$\hat{\boldsymbol{g}}_i^t$ is an unbiased estimator of the gradient $\nabla_{\boldsymbol{\mu}} \mathbb{E}_{p_{\boldsymbol{\theta}_t}}[F_i(\boldsymbol{x})]$.*

(b) *$\mathbb{E}_{\boldsymbol{z}}[\|\mathbf{\Sigma}_t \hat{\boldsymbol{g}}_i^t\|^2] \leq \frac{H^2(d+4)^2}{4\xi^2(\sum_{k=1}^t \beta_k)^2}$.*

(c) *$\mathbb{V}_{\boldsymbol{z}}[\hat{\boldsymbol{g}}_i^t] = \mathbb{E}_{\boldsymbol{z}}[\|\hat{\boldsymbol{g}}_i^t - \nabla_{\boldsymbol{\mu}} J_i(\boldsymbol{\theta}_t)\|^2] \leq \frac{H^2 C(d+4)^2}{N}$, where $C = \max(\frac{b}{\xi}, \|\mathbf{\Sigma}_0^{-1}\|_\infty)$.*

**Lemma B.5.** *Suppose $\boldsymbol{\lambda}^t = (1 - \gamma_t)\boldsymbol{\lambda}^{t-1} + \gamma_t \tilde{\boldsymbol{\lambda}}^t$, then we have $\mathbb{V}_{\boldsymbol{z}}[\boldsymbol{\lambda}^t] = \mathbb{E}_{\boldsymbol{z}}[\|\boldsymbol{\lambda}^t - \mathbb{E}_{\boldsymbol{z}}[\boldsymbol{\lambda}^t]\|^2] \leq 2\gamma_t^2$.*

**Lemma B.6.** *Suppose assumption 4.1 holds, if $\hat{\boldsymbol{g}}_1^t, \ldots, \hat{\boldsymbol{g}}_m^t$ are unbiased estimates of $\nabla_{\boldsymbol{\mu}} J_1(\boldsymbol{\theta}_t), \ldots, \nabla_{\boldsymbol{\mu}} J_m(\boldsymbol{\theta}_t)$. Further suppose that each gradient variance is bounded by $\mathbb{V}_{\boldsymbol{z}}[\hat{\boldsymbol{g}}_i^t] = \mathbb{E}_{\boldsymbol{z}}[\|\hat{\boldsymbol{g}}_i^t - \nabla_{\boldsymbol{\mu}} J_i(\boldsymbol{\theta}_t)\|^2] \leq \delta, i = 1, \ldots, m$ and let $\mathbb{V}_{\boldsymbol{z}}[\boldsymbol{\lambda}^t] = \mathbb{E}_{\boldsymbol{z}}[\|\boldsymbol{\lambda}^t - \mathbb{E}_{\boldsymbol{z}}[\boldsymbol{\lambda}^t]\|^2]$. Then for any gradient descent algorithm updated with composite gradient $\boldsymbol{q}_t = -\sum_{i=1}^m \lambda_i^t \hat{\boldsymbol{g}}_i^t$ with $\boldsymbol{\lambda}^t \in \Delta^{m-1}$, we have following inequality in $t$-th iteration,*

(a) *$\|\mathbb{E}_{\boldsymbol{z}}[-\boldsymbol{q}_t] - \mathbb{E}_{\boldsymbol{z}}[\sum_{i=1}^m \lambda_i^t \nabla_{\boldsymbol{\mu}} J_i(\boldsymbol{\theta}_t)]\|^2 \leq \mathbb{V}_{\boldsymbol{z}}[\boldsymbol{\lambda}^t] \sum_{i=1}^m \mathbb{V}_{\boldsymbol{z}}[\hat{\boldsymbol{g}}_i^t]$.*

(b) *$\mathbb{E}_{\boldsymbol{z}}[(\sum_{i=1}^m \lambda_i^t \nabla_{\boldsymbol{\mu}} J_i(\boldsymbol{\theta}_t))^\top \boldsymbol{q}_t] \leq 2H\sqrt{\mathbb{V}_{\boldsymbol{z}}[\boldsymbol{\lambda}^t] \sum_{i=1}^m \mathbb{V}_{\boldsymbol{z}}[\hat{\boldsymbol{g}}_i^t]} - \mathbb{E}_{\boldsymbol{z}}[\|\sum_{i=1}^m \lambda_i^t \nabla_{\boldsymbol{\mu}} J_i(\boldsymbol{\theta}_t)\|^2]$.*

(c) *$\mathbb{E}_{\boldsymbol{z}}[\|\boldsymbol{q}_t\|^2] - \|\sum_{i=1}^m \lambda_i^t \nabla_{\boldsymbol{\mu}} J_i(\boldsymbol{\theta}_t)\|^2] \leq \sum_{i=1}^m \mathbb{V}_{\boldsymbol{z}}[\hat{\boldsymbol{g}}_i^t] + 4H\sqrt{\mathbb{V}_{\boldsymbol{z}}[\boldsymbol{\lambda}^t] \sum_{i=1}^m \mathbb{V}_{\boldsymbol{z}}[\hat{\boldsymbol{g}}_i^t]}$.*

## C  PROOF OF THE RESULT IN SECTION 4

In this section, we provide the proof of the result in Section 4.

Theorem 4.2 can be directly obtained by Lemma B.3 (a).

### C.1  PROOF OF THE PROPOSITION 4.3

From the definition of $J_i(\boldsymbol{\mu}, \mathbf{\Sigma})$, we know that $F_i(\boldsymbol{\mu}^*) = J_i(\boldsymbol{\mu}^*, \mathbf{0})$. Note that $F_i(\boldsymbol{x})$ is a convex function, we have that

$$F_i(\boldsymbol{\mu}) = F_i(\mathbb{E}_{\boldsymbol{x} \sim \mathcal{N}(\boldsymbol{\mu}, \mathbf{\Sigma})}[\boldsymbol{x}]) \leq \mathbb{E}_{\boldsymbol{x} \sim \mathcal{N}(\boldsymbol{\mu}, \mathbf{\Sigma})}[F_i(\boldsymbol{x})] = J_i(\boldsymbol{\mu}, \mathbf{\Sigma}). \tag{44}$$

It follows that

$$F_i(\boldsymbol{\mu}) - F_i(\boldsymbol{\mu}^*) \leq J_i(\boldsymbol{\mu}, \mathbf{\Sigma}) - J_i(\boldsymbol{\mu}^*, \mathbf{0}). \tag{45}$$

Then we have

$$\sum_{i=1}^m \lambda_i(F_i(\boldsymbol{\mu}) - F_i(\boldsymbol{\mu}^*)) \leq \sum_{i=1}^m \lambda_i(J_i(\boldsymbol{\mu}, \mathbf{\Sigma}) - J_i(\boldsymbol{\mu}^*, \mathbf{0})), \tag{46}$$

where we reach the conclusion.

### C.2  PROOF OF THE PROPOSITION 4.5

Note that $\|\sum_{i=1}^m \lambda_i \nabla_{\boldsymbol{\mu}^*} J_i(\boldsymbol{\mu}^*)\| = 0$, we have

$$\|\sum_{i=1}^m \lambda_i \nabla F_i(\boldsymbol{\mu}^*)\|^2 = \left\|\sum_{i=1}^m \lambda_i \nabla F_i(\boldsymbol{\mu}^*) - \sum_{i=1}^m \lambda_i \nabla_{\boldsymbol{\mu}^*} J_i(\boldsymbol{\mu}^*) + \sum_{i=1}^m \lambda_i \nabla_{\boldsymbol{\mu}^*} J_i(\boldsymbol{\mu}^*)\right\|^2 \tag{47}$$

$$= \left\|\sum_{i=1}^m \lambda_i \nabla F_i(\boldsymbol{\mu}^*) - \sum_{i=1}^m \lambda_i \nabla_{\boldsymbol{\mu}^*} J_i(\boldsymbol{\mu}^*)\right\|^2. \tag{48}$$

It follows that

$$\| \sum_{i=1}^{m} \lambda_i \nabla F_i(\boldsymbol{\mu}^*) \|_2^2 \leq \sum_{i=1}^{m} \lambda_i \left\| \nabla F_i(\boldsymbol{\mu}^*) - \nabla_{\boldsymbol{\mu}^*} J_i(\boldsymbol{\mu}^*) \right\|^2 \tag{49}$$

$$= \sum_{i=1}^{m} \lambda_i \left\| \nabla F_i(\boldsymbol{\mu}^*) - \mathbb{E}_{\boldsymbol{x} \sim \mathcal{N}(\boldsymbol{\mu}^*, \boldsymbol{\sigma})} \nabla F_i(\boldsymbol{x}) \right\|^2 \tag{50}$$

$$\leq \sum_{i=1}^{m} \lambda_i \mathbb{E}_{\boldsymbol{x} \sim \mathcal{N}(\boldsymbol{\mu}^*, \boldsymbol{\sigma})} \| \nabla F_i(\boldsymbol{x}) - \nabla F_i(\boldsymbol{\mu}^*) \|^2 \tag{51}$$

$$\leq L_F^2 \mathbb{E}_{\boldsymbol{x} \sim \mathcal{N}(\boldsymbol{\mu}^*, \boldsymbol{\sigma})} \| \boldsymbol{x} - \boldsymbol{\mu}^* \|^2 \tag{52}$$

$$= L_F^2 \| \text{diag}(\boldsymbol{\Sigma}) \|_1, \tag{53}$$

where the equality in Eq. (50) is due to $\nabla_{\boldsymbol{\mu}^*} J_i(\boldsymbol{\mu}^*) = \mathbb{E}_{\boldsymbol{x} \sim \mathcal{N}(\boldsymbol{\mu}^*, \boldsymbol{\sigma})} \nabla F_i(\boldsymbol{x})$ in Rezende et al. (2014).

## C.3    PROOF OF THEOREM 4.4

We denote $\boldsymbol{q}_t = -\sum_{i=1}^{m} \lambda_i^t \hat{\boldsymbol{g}}_i^t$, then the update rule of $\boldsymbol{\mu}$ can be represented as $\boldsymbol{\mu}_{t+1} = \boldsymbol{\mu}_t + \beta_t \boldsymbol{\Sigma}_t \boldsymbol{q}_t$.
According to assumption 4.1, the function $J_i(\boldsymbol{\theta})$ is $L$-smooth w.r.t $\{\boldsymbol{\mu}, \boldsymbol{\Sigma}\}$, then we have

$$\lambda_i^t J_i(\boldsymbol{\mu}_{t+1}, \boldsymbol{\Sigma}_t) \leq \lambda_i^t \left( J_i(\boldsymbol{\mu}_t, \boldsymbol{\Sigma}_t) + \beta_t \nabla_{\boldsymbol{\mu}} J_i(\boldsymbol{\theta}_t, \boldsymbol{\Sigma}_t)^\top \boldsymbol{\Sigma}_t \boldsymbol{q}_t + \frac{L\beta_t^2}{2} \|\boldsymbol{\Sigma}_t \boldsymbol{q}_t\|^2 \right). \tag{54}$$

Since $F_i(\boldsymbol{x})$ is convex function, we have $J_i(\boldsymbol{\theta})$ is convex w.r.t $\boldsymbol{\theta} = \{\boldsymbol{\mu}, \boldsymbol{\Sigma}^{\frac{1}{2}}\}$ by Lemma B.2, together with $J_i(\boldsymbol{\theta})$ is $c$-strongly convex w.r.t $\boldsymbol{\mu}$ we obtain

$$J_i(\boldsymbol{\theta}_t) \leq J_i(\boldsymbol{\mu}^*, 0) + \nabla_{\boldsymbol{\mu}} J_i(\boldsymbol{\theta}_t)^\top (\boldsymbol{\mu}_t - \boldsymbol{\mu}^*) + \nabla_{\boldsymbol{\Sigma}^{\frac{1}{2}}} J_i(\boldsymbol{\theta}_t)^\top \boldsymbol{\Sigma}_t^{\frac{1}{2}} - \frac{c}{2} \|\boldsymbol{\mu}_t - \boldsymbol{\mu}^*\|^2. \tag{55}$$

Note that $\nabla_{\boldsymbol{\Sigma}^{\frac{1}{2}}} J(\boldsymbol{\theta}_t) = \boldsymbol{\Sigma}_t^{\frac{1}{2}} \nabla_{\boldsymbol{\Sigma}} J(\boldsymbol{\theta}_t) + \nabla_{\boldsymbol{\Sigma}} J(\boldsymbol{\theta}_t) \boldsymbol{\Sigma}_t^{\frac{1}{2}}$, we have

$$J_i(\boldsymbol{\theta}_t) \leq J_i(\boldsymbol{\mu}^*, 0) + \nabla_{\boldsymbol{\mu}} J_i(\boldsymbol{\theta}_t)^\top (\boldsymbol{\mu}_t - \boldsymbol{\mu}^*) + 2\nabla_{\boldsymbol{\Sigma}} J_i(\boldsymbol{\theta}_t) \boldsymbol{\Sigma}_t - \frac{c}{2} \|\boldsymbol{\mu}_t - \boldsymbol{\mu}^*\|^2. \tag{56}$$

Substituting Eq. (56) into Eq. (54), we have

$$\lambda_i^t J_i(\boldsymbol{\mu}_{t+1}, \boldsymbol{\Sigma}_t) \leq \lambda_i^t J_i(\boldsymbol{\mu}^*, 0) + \lambda_i^t \nabla_{\boldsymbol{\mu}} J_i(\boldsymbol{\theta}_t)^\top (\boldsymbol{\mu}_t - \boldsymbol{\mu}^*) + 2\lambda_i^t \nabla_{\boldsymbol{\Sigma}} J_i(\boldsymbol{\theta}_t)^\top \boldsymbol{\Sigma}_t$$
$$+ \beta_t \lambda_i^t \nabla_{\boldsymbol{\mu}} J_i(\boldsymbol{\theta}_t, \boldsymbol{\Sigma}_t)^\top \boldsymbol{\Sigma}_t \boldsymbol{q}_t + \frac{L\lambda_i^t \beta_t^2}{2} \|\boldsymbol{\Sigma}_t \boldsymbol{q}_t\|^2 - \frac{c}{2} \|\boldsymbol{\mu}_t - \boldsymbol{\mu}^*\|^2. \tag{57}$$

Let $A_t = \sum_{i=1}^{m} \lambda_i^t \big( J_i(\boldsymbol{\mu}_{t+1}, \boldsymbol{\Sigma}_t) - J_i(\boldsymbol{\mu}^*, 0) \big)$ and $\beta_t \leq \frac{1}{L}$, we have

$$\mathbb{E}_{\boldsymbol{z}}[A_t] \leq \mathbb{E}_{\boldsymbol{z}}[\sum_{i=1}^{m} \lambda_i^t \nabla_{\boldsymbol{\mu}} J_i(\boldsymbol{\theta}_t)^\top (\boldsymbol{\mu}_t - \boldsymbol{\mu}^*)] + \beta_t \mathbb{E}_{\boldsymbol{z}}[\sum_{i=1}^{m} \lambda_i^t \nabla_{\boldsymbol{\mu}} J_i(\boldsymbol{\theta}_t, \boldsymbol{\Sigma}_t)^\top \boldsymbol{\Sigma}_t \boldsymbol{q}_t]$$
$$+ \frac{\beta_t}{2} \mathbb{E}_{\boldsymbol{z}}[\|\boldsymbol{\Sigma}_t \boldsymbol{q}_t\|^2] + \mathbb{E}_{\boldsymbol{z}}[\sum_{i=1}^{m} \lambda_i^t \nabla_{\boldsymbol{\Sigma}} J_i(\boldsymbol{\theta}_t)^\top \boldsymbol{\Sigma}_t] - \frac{c}{2} \|\boldsymbol{\mu}_t - \boldsymbol{\mu}^*\|^2. \tag{58}$$

Note that

$$\|\boldsymbol{\mu}_t - \boldsymbol{\mu}^*\|_{\boldsymbol{\Sigma}_t^{-1}}^2 - \|\boldsymbol{\mu}_{t+1} - \boldsymbol{\mu}^*\|_{\boldsymbol{\Sigma}_t^{-1}}^2 \tag{59}$$

$$= \|\boldsymbol{\mu}_t - \boldsymbol{\mu}^*\|_{\boldsymbol{\Sigma}_t^{-1}}^2 - \|\boldsymbol{\mu}_t + \beta_t \boldsymbol{\Sigma}_t \boldsymbol{q}_t - \boldsymbol{\mu}^*\|_{\boldsymbol{\Sigma}_t^{-1}}^2 \tag{60}$$

$$= \|\boldsymbol{\mu}_t - \boldsymbol{\mu}^*\|_{\boldsymbol{\Sigma}_t^{-1}}^2 - \left( \|\boldsymbol{\mu}_t - \boldsymbol{\mu}^*\|_{\boldsymbol{\Sigma}_t^{-1}}^2 + \beta_t \langle \boldsymbol{\mu}_t - \boldsymbol{\mu}^*, \boldsymbol{q}_t \rangle + \beta_t^2 \langle \boldsymbol{\Sigma}_t \boldsymbol{q}_t, \boldsymbol{q}_t \rangle \right) \tag{61}$$

$$= -2\beta_t \boldsymbol{q}_t^\top (\boldsymbol{\mu}_t - \boldsymbol{\mu}^*) - \beta_t^2 (\boldsymbol{\Sigma}_t \boldsymbol{q}_t)^\top \boldsymbol{q}_t. \tag{62}$$

Therefore we have

$$-\boldsymbol{q}_t^\top (\boldsymbol{\mu}_t - \boldsymbol{\mu}^*) = \frac{1}{2\beta_t} \left( \|\boldsymbol{\mu}_t - \boldsymbol{\mu}^*\|_{\boldsymbol{\Sigma}_t^{-1}}^2 - \|\boldsymbol{\mu}_{t+1} - \boldsymbol{\mu}^*\|_{\boldsymbol{\Sigma}_t^{-1}}^2 \right) + \frac{\beta_t}{2} (\boldsymbol{\Sigma}_t \boldsymbol{q}_t)^\top \boldsymbol{q}_t \tag{63}$$

$$\leq \frac{1}{2\beta_t} \left( \|\boldsymbol{\mu}_t - \boldsymbol{\mu}^*\|_{\boldsymbol{\Sigma}_t^{-1}}^2 - \|\boldsymbol{\mu}_{t+1} - \boldsymbol{\mu}^*\|_{\boldsymbol{\Sigma}_t^{-1}}^2 \right) + \beta_t \|\boldsymbol{\Sigma}_t\|_F \|\boldsymbol{q}_t\|^2, \tag{64}$$

where the inequality is due to $\beta_t \geq 0$ and Lemma B.1. Note that we have

$$\mathbb{E}_{\boldsymbol{z}}\big[(\sum_{i=1}^{m} \lambda_i^t \nabla_{\boldsymbol{\mu}} J_i(\boldsymbol{\theta}_t) + \boldsymbol{q}_t)^\top (\boldsymbol{\mu}_t - \boldsymbol{\mu}^*)\big] \leq \|\boldsymbol{\mu}_t - \boldsymbol{\mu}^*\| \sqrt{\mathbb{E}_{\boldsymbol{z}}[\|\sum_{i=1}^{m} \lambda_i^t \nabla_{\boldsymbol{\mu}} J_i(\boldsymbol{\theta}_t) + \boldsymbol{q}_t\|^2]} \quad (65)$$

$$\leq D \sqrt{\mathbb{E}_{\boldsymbol{z}}[\|\sum_{i=1}^{m} \lambda_i^t \nabla_{\boldsymbol{\mu}} J_i(\boldsymbol{\theta}_t) + \boldsymbol{q}_t\|^2]} \quad (66)$$

$$\leq D \sqrt{\mathbb{V}_{\boldsymbol{z}}[\boldsymbol{\lambda}^t] \sum_{i=1}^{m} \mathbb{V}_{\boldsymbol{z}}[\hat{\boldsymbol{g}}_i^t]}, \quad (67)$$

where the first inequality is due to the Cauchy-Schwarz inequality, the second inequality is due to $\|\boldsymbol{\mu}_t - \boldsymbol{\mu}^*\| \leq D$ and the last inequality is due to Lemma B.6 (a). Then we have

$$\mathbb{E}_{\boldsymbol{z}}\big[\sum_{i=1}^{m} \lambda_i^t \nabla_{\boldsymbol{\mu}} J_i(\boldsymbol{\theta}_t)^\top (\boldsymbol{\mu}_t - \boldsymbol{\mu}^*)\big] \quad (68)$$

$$= \mathbb{E}_{\boldsymbol{z}}\big[-\boldsymbol{q}_t^\top (\boldsymbol{\mu}_t - \boldsymbol{\mu}^*) + (\sum_{i=1}^{m} \lambda_i^t \nabla_{\boldsymbol{\mu}} J_i(\boldsymbol{\theta}_t) + \boldsymbol{q}_t)^\top (\boldsymbol{\mu}_t - \boldsymbol{\mu}^*)\big] \quad (69)$$

$$\leq \frac{1}{2\beta_t} \mathbb{E}_{\boldsymbol{z}}[\|\boldsymbol{\mu}_t - \boldsymbol{\mu}^*\|_{\boldsymbol{\Sigma}_t^{-1}}^2 - \|\boldsymbol{\mu}_{t+1} - \boldsymbol{\mu}^*\|_{\boldsymbol{\Sigma}_t^{-1}}^2] + \beta_t \|\boldsymbol{\Sigma}_t\|_F \mathbb{E}_{\boldsymbol{z}}\|\boldsymbol{q}_t\|^2] + D\sqrt{\mathbb{V}_{\boldsymbol{z}}[\boldsymbol{\lambda}^t] \sum_{i=1}^{m} \mathbb{V}_{\boldsymbol{z}}(g_i^t)}, \quad (70)$$

where the inequality is due to Eq. (64) and Eq. (67). Note that

$$\mathbb{E}_{\boldsymbol{z}}\big[\sum_{i=1}^{m} \lambda_i^t \nabla_{\boldsymbol{\Sigma}} J_i(\boldsymbol{\theta}_t)^\top \boldsymbol{\Sigma}_t\big] \leq \mathbb{E}_{\boldsymbol{z}}\big[\|\sum_{i=1}^{m} \lambda_i^t \nabla_{\boldsymbol{\Sigma}} J_i(\boldsymbol{\theta}_t)\|\big] \|\boldsymbol{\Sigma}_t\|_F \leq H \|\boldsymbol{\Sigma}_t\|_F, \quad (71)$$

where the first inequality is due to Lemma B.1 and the second inequality is due to the Lipschitz continuous assumption of the function $J_i(\boldsymbol{\theta})$. By using Lemma B.1 and Lemma B.6 (b), we further have

$$\mathbb{E}_{\boldsymbol{z}}\big[\sum_{i=1}^{m} \lambda_i^t \nabla_{\boldsymbol{\mu}} J_i(\boldsymbol{\theta}_t, \boldsymbol{\Sigma}_t)^\top \boldsymbol{\Sigma}_t \boldsymbol{q}_t\big] \leq \|\boldsymbol{\Sigma}_t\|_F \left(2H\sqrt{\mathbb{V}_{\boldsymbol{z}}[\boldsymbol{\lambda}^t] \sum_{i=1}^{m} \mathbb{V}_{\boldsymbol{z}}[\hat{\boldsymbol{g}}_i^t]} - \mathbb{E}_{\boldsymbol{z}}\big[\sum_{i=1}^{m} \lambda_i^t \nabla_{\boldsymbol{\mu}} J_i(\boldsymbol{\theta}_t)\|^2\big]\right). \quad (72)$$

Then substituting Eqs. (70) (71) (72) into Eq. (58) and multiplying $\beta_t$ on both sides of the inequality, we have

$$\beta_t \mathbb{E}_{\boldsymbol{z}}[A_t] \leq \frac{1}{2} \mathbb{E}_{\boldsymbol{z}}[\|\boldsymbol{\mu}_t - \boldsymbol{\mu}^*\|_{\boldsymbol{\Sigma}_t^{-1}}^2 - \|\boldsymbol{\mu}_{t+1} - \boldsymbol{\mu}^*\|_{\boldsymbol{\Sigma}_t^{-1}}^2] - \frac{c\beta_t}{2}\|\boldsymbol{\mu}_t - \boldsymbol{\mu}^*\|^2 + \beta_t^2 H\|\boldsymbol{\Sigma}_t\|_F$$

$$+ (2H\beta_t^2\|\boldsymbol{\Sigma}_t\|_F + \beta_t D)\sqrt{\mathbb{V}_{\boldsymbol{z}}[\boldsymbol{\lambda}^t] \sum_{i=1}^{m} \mathbb{V}_{\boldsymbol{z}}[\hat{\boldsymbol{g}}_i^t]} + \frac{\beta_t^2}{2}\mathbb{E}_{\boldsymbol{z}}[\|\boldsymbol{\Sigma}_t \boldsymbol{q}_t\|^2]$$

$$+ \beta_t^2\|\boldsymbol{\Sigma}_t\|_F \mathbb{E}_{\boldsymbol{z}}\|\boldsymbol{q}_t\|^2] - \beta_t^2\|\boldsymbol{\Sigma}_t\|_F \mathbb{E}_{\boldsymbol{z}}\big[\sum_{i=1}^{m} \lambda_i^t \nabla_{\boldsymbol{\mu}} J_i(\boldsymbol{\theta}_t)\|^2\big] \quad (73)$$

$$\leq \frac{1}{2} \mathbb{E}_{\boldsymbol{z}}[\|\boldsymbol{\mu}_t - \boldsymbol{\mu}^*\|_{\boldsymbol{\Sigma}_t^{-1}}^2 - \|\boldsymbol{\mu}_{t+1} - \boldsymbol{\mu}^*\|_{\boldsymbol{\Sigma}_t^{-1}}^2] - \frac{c\beta_t}{2}\|\boldsymbol{\mu}_t - \boldsymbol{\mu}^*\|^2 + \beta_t^2 H\|\boldsymbol{\Sigma}_t\|_F$$

$$+ (2H\beta_t^2\|\boldsymbol{\Sigma}_t\|_F + \beta_t D)\sqrt{\mathbb{V}_{\boldsymbol{z}}[\boldsymbol{\lambda}^t] \sum_{i=1}^{m} \mathbb{V}_{\boldsymbol{z}}[\hat{\boldsymbol{g}}_i^t]} + \frac{\beta_t^2 H^2 (d+4)^2}{8\xi^2 (\sum_{k=1}^{t} \beta_k)^2}$$

$$+ \beta_t^2\|\boldsymbol{\Sigma}_t\|_F \left(\sum_{i=1}^{m} \mathbb{V}_{\boldsymbol{z}}[\hat{\boldsymbol{g}}_i^t] + 4H\sqrt{\mathbb{V}_{\boldsymbol{z}}[\boldsymbol{\lambda}^t] \sum_{i=1}^{m} \mathbb{V}_{\boldsymbol{z}}[\hat{\boldsymbol{g}}_i^t]}\right), \quad (74)$$

where the second inequality is due to Lemma B.4 (b) and Lemma B.6 (c). We further obtain that

$$\sum_{t=0}^{T-1} \left[ \frac{1}{2} \mathbb{E}_{\boldsymbol{z}}[\|\boldsymbol{\mu}_t - \boldsymbol{\mu}^*\|^2_{\boldsymbol{\Sigma}_t^{-1}} - \|\boldsymbol{\mu}_{t+1} - \boldsymbol{\mu}^*\|^2_{\boldsymbol{\Sigma}_t^{-1}}] - \frac{c\beta_t}{2} \|\boldsymbol{\mu}_t - \boldsymbol{\mu}^*\|^2 \right] \tag{75}$$

$$\leq \frac{1}{2} \sum_{t=0}^{T-1} \left[ \|\boldsymbol{\mu}_t - \boldsymbol{\mu}^*\|^2_{\boldsymbol{\Sigma}_t^{-1}} - \|\boldsymbol{\mu}_{t-1} - \boldsymbol{\mu}^*\|^2_{\boldsymbol{\Sigma}_{t-1}^{-1}} - \frac{c\beta_t}{2} \|\boldsymbol{\mu}_t - \boldsymbol{\mu}^*\|^2 \right]$$

$$+ \frac{1}{2} \left[ \|\boldsymbol{\mu}_0 - \boldsymbol{\mu}^*\|^2_{\boldsymbol{\Sigma}_0^{-1}} - \|\boldsymbol{\mu}_T - \boldsymbol{\mu}^*\|^2_{\boldsymbol{\Sigma}_{T-1}^{-1}} \right] \tag{76}$$

$$\leq \frac{1}{2} \sum_{t=0}^{T-1} \left[ \|\boldsymbol{\mu}_t - \boldsymbol{\mu}^*\|^2_{2\beta_t \sum_{i=1}^m \lambda_i^t \hat{G}_i^t} - \frac{c\beta_t}{2} \|\boldsymbol{\mu}_t - \boldsymbol{\mu}^*\|^2 \right] + \|\boldsymbol{\Sigma}_0^{-1}\|_F D^2 \tag{77}$$

$$\leq \frac{1}{2} \sum_{t=0}^{T-1} \left[ \frac{c\beta_t}{2} \|\boldsymbol{\mu}_t - \boldsymbol{\mu}^*\|^2 - \frac{c\beta_t}{2} \|\boldsymbol{\mu}_t - \boldsymbol{\mu}^*\|^2 \right] + \|\boldsymbol{\Sigma}_0^{-1}\|_F D^2 \tag{78}$$

$$= \|\boldsymbol{\Sigma}_0^{-1}\|_F D^2, \tag{79}$$

where the second inequality is due to the update rule of $\boldsymbol{\Sigma}_t$ and $\|\boldsymbol{\mu}_t - \boldsymbol{\mu}^*\| \leq D$, and the third inequality is due to Cauchy-Schwarz inequality and $\hat{G}_i \preceq \frac{c}{4} \boldsymbol{I}$. Let $C = \max(\frac{c}{4\xi}, \|\boldsymbol{\Sigma}_0^{-1}\|_\infty)$, we have

$$\sum_{t=0}^{T-1} \beta_t \mathbb{E}_{\boldsymbol{z}}[A_t] \leq \|\boldsymbol{\Sigma}_0^{-1}\|_F D^2 + \sum_{t=0}^{T-1} \left( \frac{\beta_t^2 H \sqrt{d}}{2\xi \sum_{k=1}^t \beta_k} + \frac{\beta_t^2 H^2 (d+4)^2}{8\xi^2 (\sum_{k=1}^t \beta_k)^2} \right.$$

$$\left. + (6H\beta_t^2 \|\boldsymbol{\Sigma}_t\|_F + \beta_t D) \sqrt{\mathbb{V}_{\boldsymbol{z}}[\boldsymbol{\lambda}^t] \sum_{i=1}^m \mathbb{V}_{\boldsymbol{z}}[\hat{\boldsymbol{g}}_i^t] + \beta_t^2 \|\boldsymbol{\Sigma}_t\|_F \sum_{i=1}^m \mathbb{V}_{\boldsymbol{z}}[\hat{\boldsymbol{g}}_i^t]} \right) \tag{80}$$

$$\leq \|\boldsymbol{\Sigma}_0^{-1}\|_F D^2 + \sum_{t=0}^{T-1} \left( \frac{\beta_t^2 H \sqrt{d}}{2\xi \sum_{k=1}^t \beta_k} + \frac{\beta_t^2 H^2 (d+4)^2}{8\xi^2 (\sum_{k=1}^t \beta_k)^2} \right.$$

$$\left. + (6H\beta_t^2 \sqrt{d}R + \beta_t D)\gamma_t \sqrt{2 \sum_{i=1}^m \mathbb{V}_{\boldsymbol{z}}[\hat{\boldsymbol{g}}_i^t] + \frac{\beta_t^2 \sqrt{d} \sum_{i=1}^m \mathbb{V}_{\boldsymbol{z}}[\hat{\boldsymbol{g}}_i^t]}{2\xi \sum_{k=1}^t \beta_k}} \right) \tag{81}$$

$$\leq \|\boldsymbol{\Sigma}_0^{-1}\|_F D^2 + \sum_{t=0}^{T-1} \left( \frac{\beta_t^2 H \sqrt{d}}{2\xi \sum_{k=1}^t \beta_k} + \frac{\beta_t^2 H^2 (d+4)^2}{8\xi^2 (\sum_{k=1}^t \beta_k)^2} \right.$$

$$\left. + H\gamma_t \beta_t (d+4)(6H\beta_t \sqrt{d}R + D) \sqrt{\frac{2Cm}{N}} + \frac{\beta_t^2 \sqrt{d} H^2 C (d+4)^2 m}{2N\xi \sum_{k=1}^t \beta_k} \right), \tag{82}$$

where the first inequality is due to Eq. (79) and Lemma B.3 (a), the second inequality is due to Lemma B.3 (b) and Lemma B.5, and the third inequality is due to Lemma B.4 (c).

Therefore, we have

$$\frac{1}{T} \sum_{t=0}^{T-1} \mathbb{E}_{\boldsymbol{z}}[A_t] \leq \frac{\|\boldsymbol{\Sigma}_0^{-1}\|_F D^2}{T\beta_t} + \frac{1}{T} \sum_{t=0}^{T-1} \left( \frac{\beta_t H \sqrt{d}}{2\xi \sum_{k=1}^t \beta_k} + \frac{\beta_t H^2 (d+4)^2}{8\xi^2 (\sum_{k=1}^t \beta_k)^2} \right.$$

$$\left. + H\gamma_t (d+4)(6H\beta_t \sqrt{d}R + D) \sqrt{\frac{2Cm}{N}} + \frac{\beta_t \sqrt{d} H^2 C (d+4)^2 m}{2N\xi \sum_{k=1}^t \beta_k} \right). \tag{83}$$

Let $\beta_t = \beta$ and $\gamma_t = \gamma$. Since we have $\sum_{t=1}^T \frac{1}{t} \leq 1 + \log(T)$, we obtain

$$\frac{1}{T} \sum_{t=0}^{T-1} \mathbb{E}_{\boldsymbol{z}} \left[ \sum_{i=1}^m \lambda_i^t (J_i(\boldsymbol{\mu}_{t+1}, \boldsymbol{\Sigma}_t) - J_i(\boldsymbol{\mu}^*, 0)) \right] = \frac{1}{T} \sum_{t=0}^{T-1} \mathbb{E}_{\boldsymbol{z}}[A_t] = \mathcal{O}\left( \frac{1}{\beta T} + \frac{\log T}{T} + \gamma \right), \tag{84}$$

where we reach the conclusion.

## C.4 Proof of Theorem 4.6

We denote $\boldsymbol{q}_t = -\sum_{i=1}^m \lambda_i^t \hat{\boldsymbol{g}}_i^t$, then the update rule of $\boldsymbol{\mu}$ can be represented as $\boldsymbol{\mu}_{t+1} = \boldsymbol{\mu}_t + \beta_t \boldsymbol{\Sigma}_t \boldsymbol{q}_t$. According to assumption 4.1, the function $J_i(\boldsymbol{\theta})$ is $L$-smooth w.r.t $\{\boldsymbol{\mu}, \boldsymbol{\Sigma}\}$, then we have

$$\lambda_i^t J_i(\boldsymbol{\mu}_{t+1}, \boldsymbol{\Sigma}_t) \le \lambda_i^t \left( J_i(\boldsymbol{\mu}_t, \boldsymbol{\Sigma}_t) + \beta_t \nabla_{\boldsymbol{\mu}} J_i(\boldsymbol{\theta}_t, \boldsymbol{\Sigma}_t)^\top \boldsymbol{\Sigma}_t \boldsymbol{q}_t + \frac{L\beta_t^2}{2} \|\boldsymbol{\Sigma}_t \boldsymbol{q}_t\|^2 \right). \tag{85}$$

Let $B_t = \mathbb{E}_{\boldsymbol{z}}[\sum_{i=1}^m \lambda_i^t (J_i(\boldsymbol{\mu}_{t+1}, \boldsymbol{\Sigma}_t) - J_i(\boldsymbol{\mu}_t, \boldsymbol{\Sigma}_t))]$, then we have

$$B_t \le \beta_t \|\boldsymbol{\Sigma}_t\|_F \mathbb{E}_{\boldsymbol{z}}[(\sum_{i=1}^m \lambda_i^t \nabla_{\boldsymbol{\mu}} J_i(\boldsymbol{\theta}_t))^\top \boldsymbol{q}_t] + \frac{L\beta_t^2 \|\boldsymbol{\Sigma}_t\|_F^2}{2} \mathbb{E}_{\boldsymbol{z}}[\|\boldsymbol{q}_t\|^2] \tag{86}$$

$$\le 2H\beta_t\sqrt{d}R\sqrt{\mathbb{V}_{\boldsymbol{z}}[\boldsymbol{\lambda}^t] \sum_{i=1}^m \mathbb{V}_{\boldsymbol{z}}[\hat{\boldsymbol{g}}_i^t]} - \beta_t\sqrt{d}R\mathbb{E}_{\boldsymbol{z}}[\| \sum_{i=1}^m \lambda_i^t \nabla_{\boldsymbol{\mu}} J_i(\boldsymbol{\theta}_t)\|^2]$$

$$+ \frac{L\beta_t^2 dR^2}{2}\mathbb{E}_{\boldsymbol{z}}[\|\boldsymbol{q}_t\|^2] \tag{87}$$

$$\le (2H\beta_t\sqrt{d}R + 2HL\beta_t^2 dR^2)\sqrt{\mathbb{V}_{\boldsymbol{z}}[\boldsymbol{\lambda}^t] \sum_{i=1}^m \mathbb{V}_{\boldsymbol{z}}[\hat{\boldsymbol{g}}_i^t]} + \frac{L\beta_t^2 dR^2}{2}\sum_{i=1}^m \mathbb{V}_{\boldsymbol{z}}[\hat{\boldsymbol{g}}_i^t]$$

$$+ \frac{L\beta_t^2 dR^2 - 2\beta_t\sqrt{d}R}{2}\mathbb{E}_{\boldsymbol{z}}[\| \sum_{i=1}^m \lambda_i^t \nabla_{\boldsymbol{\mu}} J_i(\boldsymbol{\theta}_t)\|^2], \tag{88}$$

where the first inequality is due to Lemma B.1, the second inequality is due to $\|\boldsymbol{\Sigma}_t\|_F \le \|\boldsymbol{\Sigma}_0\|_F \le \sqrt{d}R$ and Lemma B.6 (b), and the last inequality is due to Lemma B.6 (c). Let $\beta_t \le \frac{1}{LR\sqrt{d}}$, and rearrange Eq. (88) we obtain

$$\frac{\beta_t\sqrt{d}R}{2}\mathbb{E}_{\boldsymbol{z}}[\| \sum_{i=1}^m \lambda_i^t \nabla_{\boldsymbol{\mu}} J_i(\boldsymbol{\theta}_t)\|^2] \le B_t + 4HR\sqrt{d}\beta_t\sqrt{\mathbb{V}_{\boldsymbol{z}}[\boldsymbol{\lambda}^t] \sum_{i=1}^m \mathbb{V}_{\boldsymbol{z}}[\hat{\boldsymbol{g}}_i^t]} + \frac{L\beta_t^2 dR^2}{2}\sum_{i=1}^m \mathbb{V}_{\boldsymbol{z}}[\hat{\boldsymbol{g}}_i^t]. \tag{89}$$

So we have

$$\beta_t \mathbb{E}_{\boldsymbol{z}}[\| \sum_{i=1}^m \lambda_i^t \nabla_{\boldsymbol{\mu}} J_i(\boldsymbol{\theta}_t)\|^2] \le \frac{2B_t}{\sqrt{d}R} + 8H\beta_t\sqrt{\mathbb{V}_{\boldsymbol{z}}[\boldsymbol{\lambda}^t] \sum_{i=1}^m \mathbb{V}_{\boldsymbol{z}}[\hat{\boldsymbol{g}}_i^t]} + \frac{L\beta_t^2\sqrt{d}R}{2}\sum_{i=1}^m \mathbb{V}_{\boldsymbol{z}}[\hat{\boldsymbol{g}}_i^t]. \tag{90}$$

Note that we have

$$\sum_{t=0}^{T-1} \sum_{i=1}^m \lambda_i^t (J_i(\boldsymbol{\theta}_{t+1}) - J_i(\boldsymbol{\theta}_t)) = \sum_{t=0}^{T-1} \sum_{i=1}^m (\lambda_i^t - \lambda_i^{t+1}) J_i(\boldsymbol{\theta}_{t+1}) + \sum_{i=1}^m (\lambda_i^{T-1} J_i(\boldsymbol{\theta}_T) - \lambda_i^0 J_i(\boldsymbol{\theta}_0)) \tag{91}$$

$$\le \sum_{t=0}^{T-1} \sum_{i=1}^m |\lambda_i^t - (1-\gamma_t)\lambda_i^t - \gamma_t \tilde{\lambda}_i^{t+1}| B + 2B \tag{92}$$

$$\le \sum_{t=0}^{T-1} \gamma_t \sum_{i=1}^m |\lambda_i^t - \tilde{\lambda}_i^{t+1}| B + 2B, \tag{93}$$

where the first inequality is due to the update rule of $\boldsymbol{\lambda}^t$ and $|J_i(\boldsymbol{\theta})| \leq B$. Then we have

$$\sum_{t=0}^{T-1} B_t = \sum_{t=0}^{T-1} \mathbb{E}_{\boldsymbol{z}}[\sum_{i=1}^{m} \lambda_i^t(J_i(\boldsymbol{\theta}_{t+1}) - J_i(\boldsymbol{\theta}_t))] + \sum_{t=0}^{T-1} \mathbb{E}_{\boldsymbol{z}}[\sum_{i=1}^{m} \lambda_i^t(J_i(\boldsymbol{\mu}_{t+1}, \boldsymbol{\Sigma}_t) - J_i(\boldsymbol{\mu}_{t+1}, \boldsymbol{\Sigma}_{t+1}))] \tag{94}$$

$$\leq \sum_{t=0}^{T-1} \gamma_t \sum_{i=1}^{m} |\lambda_i^t - \tilde{\lambda}_i^{t+1}|B + 2B + \sum_{t=0}^{T-1} \mathbb{E}_{\boldsymbol{z}}[\sum_{i=1}^{m} \lambda_i^t H\|\boldsymbol{\Sigma}_{t+1} - \boldsymbol{\Sigma}_t\|_F] \tag{95}$$

$$\leq 2mB \sum_{t=0}^{T-1} \gamma_t + 2B + \sum_{t=0}^{T-1} H\|\boldsymbol{\Sigma}_{t+1} - \boldsymbol{\Sigma}_t\|_F, \tag{96}$$

where the first inequality is due to Eq. (93) and the Lipschitz continuous assumption of the function $J_i(\boldsymbol{\theta})$. Substituting Eq. (96) into Eq. (90), we have

$$\frac{1}{T} \sum_{t=0}^{T-1} \beta_t \mathbb{E}_{\boldsymbol{z}}[\|\sum_{i=1}^{m} \lambda_i^t \nabla_{\boldsymbol{\mu}} J_i(\boldsymbol{\theta}_t)\|^2] \leq \frac{2mB \sum_{t=0}^{T-1} \gamma_t + 2B + \sum_{t=0}^{T-1} H\|\boldsymbol{\Sigma}_{t+1} - \boldsymbol{\Sigma}_t\|_F}{\sqrt{d}RT}$$

$$+ \frac{1}{T} \sum_{t=0}^{T-1} \left( 8H\beta_t \sqrt{\mathbb{V}_{\boldsymbol{z}}[\boldsymbol{\lambda}^t] \sum_{i=1}^{m} \mathbb{V}_{\boldsymbol{z}}[\hat{\boldsymbol{g}}_i^t]} + \frac{L\beta_t^2 \sqrt{d}R}{2} \sum_{i=1}^{m} \mathbb{V}_{\boldsymbol{z}}[\hat{\boldsymbol{g}}_i^t] \right). \tag{97}$$

According to Lemma B.3 (c), B.4 (c), and B.5. We know that $\|\boldsymbol{\Sigma}_{t+1} - \boldsymbol{\Sigma}_t\|_F \leq \frac{b\beta_t d^{\frac{3}{2}}}{2\xi^2(\sum_{k=1}^{t}\beta_k)^2}$, $\mathbb{V}_{\boldsymbol{z}}[\hat{\boldsymbol{g}}_i^t] \leq \frac{H^2 C(d+4)^2}{N}$, where $C = \max(\frac{b}{\xi}, \|\boldsymbol{\Sigma}_0^{-1}\|_\infty)$, and $\mathbb{V}_{\boldsymbol{z}}[\boldsymbol{\lambda}^t] \leq 2\gamma_t^2$. Then we have

$$\frac{1}{T} \sum_{t=0}^{T-1} \beta_t \mathbb{E}_{\boldsymbol{z}}[\|\sum_{i=1}^{m} \lambda_i^t \nabla_{\boldsymbol{\mu}} J_i(\boldsymbol{\theta}_t)\|^2] \leq \frac{2mB \sum_{t=0}^{T-1} \gamma_t + 2B + \sum_{t=0}^{T-1} H\frac{b\beta_t d^{\frac{3}{2}}}{2\xi^2(\sum_{k=1}^{t}\beta_k)^2}}{\sqrt{d}RT}$$

$$+ \frac{1}{T} \sum_{t=0}^{T-1} \left( 8H^2 C(d+4)\gamma_t \beta_t \sqrt{\frac{2m}{N}} + \frac{\beta_t^2 H^2 C(d+4)^2 L\sqrt{d}Rm}{2N} \right). \tag{98}$$

Let $\beta_t = \beta$ and $\gamma_t = \gamma$, we obtain

$$\frac{1}{T} \sum_{t=0}^{T-1} \mathbb{E}_{\boldsymbol{z}}[\|\sum_{i=1}^{m} \lambda_i^t \nabla_{\boldsymbol{\mu}} J_i(\boldsymbol{\theta}_t)\|^2] = \mathcal{O}\left( \frac{\gamma}{\beta} + \frac{1}{\beta T} + \gamma + \beta \right), \tag{99}$$

where we reach the conclusion.

## D PROOF OF TECHNICAL LEMMAS

In this section, we provide the proof of lemmas in Appendix B.

### D.1 PROOF OF LEMMA B.1

Since $\boldsymbol{\Sigma}$ and $\hat{\boldsymbol{\Sigma}}$ are both diagonal matrix. Denote $\boldsymbol{\sigma} = \mathrm{diag}(\boldsymbol{\Sigma})$ and $\hat{\boldsymbol{\sigma}} = \mathrm{diag}(\hat{\boldsymbol{\Sigma}})$. Then we have

$$\|\boldsymbol{\Sigma}\boldsymbol{z}\|^2 = \sum_{i=1}^{d}(\sigma_i z_i)^2 \leq \sum_{i=1}^{d}(\sigma_i)^2 \sum_{i=1}^{d}(z_i)^2 = \|\boldsymbol{\sigma}\|^2\|\boldsymbol{z}\|^2 = \|\boldsymbol{\Sigma}\|_F^2\|\boldsymbol{z}\|^2. \tag{100}$$

We further have

$$\|\boldsymbol{\Sigma}\hat{\boldsymbol{\Sigma}}\|_F^2 = \sum_{i=1}^{d}(\sigma_i \hat{\sigma}_i)^2 \leq \sum_{i=1}^{d}(\sigma_i)^2 \sum_{i=1}^{d}(\hat{\sigma}_i)^2 = \|\boldsymbol{\sigma}\|^2\|\hat{\boldsymbol{\sigma}}\|^2 = \|\boldsymbol{\Sigma}\|_F^2\|\hat{\boldsymbol{\Sigma}}\|_F^2. \tag{101}$$

Then we reach the conclusion.

### D.2 PROOF OF LEMMA B.2

For $\lambda \in [0, 1]$, we have

$$\lambda \bar{J}(\boldsymbol{\theta}_1) + (1 - \lambda)\bar{J}(\boldsymbol{\theta}_2) = \lambda \mathbb{E}_{\boldsymbol{z} \sim \mathcal{N}(\boldsymbol{0}, \boldsymbol{I})}[f(\boldsymbol{\mu}_1 + \boldsymbol{\Sigma}_1^{\frac{1}{2}}\boldsymbol{z})] + (1 - \lambda)\mathbb{E}_{\boldsymbol{z} \sim \mathcal{N}(\boldsymbol{0}, \boldsymbol{I})}[f(\boldsymbol{\mu}_2 + \boldsymbol{\Sigma}_2^{\frac{1}{2}}\boldsymbol{z})] \tag{102}$$

$$= \mathbb{E}[\lambda f(\boldsymbol{\mu}_1 + \boldsymbol{\Sigma}_1^{\frac{1}{2}}\boldsymbol{z}) + (1 - \lambda)f(\boldsymbol{\mu}_2 + \boldsymbol{\Sigma}_2^{\frac{1}{2}}\boldsymbol{z})] \tag{103}$$

$$\geq \mathbb{E}[f\left(\lambda\boldsymbol{\mu}_1 + (1 - \lambda)\boldsymbol{\mu}_2 + (\lambda\boldsymbol{\Sigma}_1^{\frac{1}{2}} + (1 - \lambda)\boldsymbol{\Sigma}_2^{\frac{1}{2}})\boldsymbol{z}\right)] \tag{104}$$

$$= \bar{J}(\lambda\boldsymbol{\theta}_1 + (1 - \lambda)\boldsymbol{\theta}_2), \tag{105}$$

where we reach the conclusion.

### D.3 PROOF OF LEMMA B.3

(a): Since we have $\boldsymbol{\Sigma}_{t+1}^{-1} = \boldsymbol{\Sigma}_t^{-1} + 2\beta_t \sum_{i=1}^m \lambda_i^t \hat{G}_i^t$ and $\boldsymbol{\lambda}^t \in \Delta^{m-1}$. We obtain

$$\boldsymbol{\Sigma}_t^{-1} + 2b\beta_t \boldsymbol{I} \succeq \boldsymbol{\Sigma}_{t+1}^{-1} \succeq \boldsymbol{\Sigma}_t^{-1} + 2\xi\beta_t \boldsymbol{I}. \tag{106}$$

Summing up it over $t = 0, \ldots, T - 1$, we have

$$\boldsymbol{\Sigma}_0^{-1} + 2b\sum_{t=1}^T \beta_t \boldsymbol{I} \succeq \boldsymbol{\Sigma}_T^{-1} \succeq \boldsymbol{\Sigma}_0^{-1} + 2\xi\sum_{t=1}^T \beta_t \boldsymbol{I}. \tag{107}$$

Therefore, we have

$$\frac{1}{2b\sum_{t=1}^T \beta_t \boldsymbol{I} + \boldsymbol{\Sigma}_0^{-1}} \preceq \boldsymbol{\Sigma}_T \preceq \frac{1}{2\xi\sum_{t=1}^T \beta_t \boldsymbol{I} + \boldsymbol{\Sigma}_0^{-1}}. \tag{108}$$

(b): We have

$$\|\boldsymbol{\Sigma}_t\|_F \leq \left\|\frac{1}{2\xi\sum_{t=1}^T \beta_t \boldsymbol{I} + \boldsymbol{\Sigma}_0^{-1}}\right\|_F \leq \left\|\frac{1}{2\xi\sum_{t=1}^T \beta_t \boldsymbol{I}}\right\|_F = \frac{\sqrt{d}}{2\xi\sum_{t=1}^T \beta_t}. \tag{109}$$

(c): We have

$$\|\boldsymbol{\Sigma}_{t+1} - \boldsymbol{\Sigma}_t\|_F = \|\frac{1}{\boldsymbol{\Sigma}_t^{-1} + 2\beta_t \sum_{i=1}^m \lambda_i^t \hat{G}_i^t} - \boldsymbol{\Sigma}_t\|_F \leq \|\frac{-2\beta_t \boldsymbol{\Sigma}_t \boldsymbol{\Sigma}_t \sum_{i=1}^m \lambda_i^t \hat{G}_i^t}{\boldsymbol{I} + 2\beta_t \boldsymbol{\Sigma}_t \sum_{i=1}^m \lambda_i^t \hat{G}_i^t}\|_F \tag{110}$$

$$\leq 2\beta_t \|\boldsymbol{\Sigma}_t\|_F^2 \|\sum_{i=1}^m \lambda_i^t \hat{G}_i^t\|_F. \tag{111}$$

Since $\|\boldsymbol{\Sigma}_t\| \leq \frac{\sqrt{d}}{2\xi\sum_{t=1}^T \beta_t}$ and $\|\sum_{i=1}^m \lambda_i^t \hat{G}_i^t\|_F \leq b\sqrt{d}$. Then we have

$$\|\boldsymbol{\Sigma}_{t+1} - \boldsymbol{\Sigma}_t\|_F \leq \frac{b\beta_t d^{\frac{3}{2}}}{2\xi^2(\sum_{t=1}^T \beta_t)^2}. \tag{112}$$

### D.4 PROOF OF LEMMA B.4

(a). We first show that $\hat{\boldsymbol{g}}_i^t$ is a unbiased estimator of $\nabla_{\boldsymbol{\mu}} \mathbb{E}_{p_{\boldsymbol{\theta}_t}}[F_i(\boldsymbol{x})]$.

$$\mathbb{E}_{\boldsymbol{z}}[\hat{\boldsymbol{g}}_i^t] = \mathbb{E}_{\boldsymbol{z}}[\boldsymbol{\Sigma}_t^{-\frac{1}{2}} \boldsymbol{z} F_i(\boldsymbol{\mu}_t + \boldsymbol{\Sigma}_t^{\frac{1}{2}}\boldsymbol{z})] - \mathbb{E}_{\boldsymbol{z}}[\boldsymbol{\Sigma}_t^{-\frac{1}{2}} \boldsymbol{z} F_i(\boldsymbol{\mu}_t)] \tag{113}$$

$$= \mathbb{E}_{\boldsymbol{z}}[\boldsymbol{\Sigma}_t^{-\frac{1}{2}} \boldsymbol{z} F_i(\boldsymbol{\mu}_t + \boldsymbol{\Sigma}_t^{\frac{1}{2}}\boldsymbol{z})] \tag{114}$$

$$= \mathbb{E}_{\boldsymbol{x} \sim \mathcal{N}(\boldsymbol{\mu}_t, \boldsymbol{\Sigma}_t)}[\boldsymbol{\Sigma}_t^{-1}(\boldsymbol{x} - \boldsymbol{\mu}_t)F_i(\boldsymbol{x})] \tag{115}$$

$$= \nabla_{\boldsymbol{\mu}} \mathbb{E}_{p_{\boldsymbol{\theta}_t}}[F_i(\boldsymbol{x})]. \tag{116}$$

(b). Since the $\boldsymbol{\sigma}$ is the diagonal elements of $\boldsymbol{\Sigma}$, then we have

$$\|\boldsymbol{\Sigma}_t \hat{\boldsymbol{g}}_{ij}^t\|^2 = \|\boldsymbol{\sigma}_t \odot \boldsymbol{\sigma}_t^{-\frac{1}{2}} \odot \boldsymbol{z}_j (F_i(\boldsymbol{\mu}_t + \boldsymbol{\sigma}_t^{\frac{1}{2}} \odot \boldsymbol{z}_j) - F_i(\boldsymbol{\mu}_t))\|^2 \tag{117}$$

$$= \|\boldsymbol{\sigma}_t^{\frac{1}{2}} \odot \boldsymbol{z}_j\|^2 (F_i(\boldsymbol{\mu}_t + \boldsymbol{\sigma}_t^{\frac{1}{2}} \odot \boldsymbol{z}_j) - F_i(\boldsymbol{\mu}_t))^2 \tag{118}$$

$$\leq \|\boldsymbol{\sigma}_t^{\frac{1}{2}} \odot \boldsymbol{z}_j\|^2 H^2 \|\boldsymbol{\sigma}_t^{\frac{1}{2}} \odot \boldsymbol{z}_j\|^2 \tag{119}$$

$$\leq H^2 \|\boldsymbol{\sigma}_t^{\frac{1}{2}}\|_\infty^2 \times \|\boldsymbol{\sigma}_t^{\frac{1}{2}}\|_\infty^2 \times \|\boldsymbol{z}_j\|^4 = H^2 \|\boldsymbol{\sigma}_t\|_\infty^2 \|\boldsymbol{z}_j\|^4. \tag{120}$$

It follows that

$$\|\boldsymbol{\Sigma}_t \hat{\boldsymbol{g}}_i^t\|_2^2 = \left\| \frac{1}{N} \sum_{j=1}^N \boldsymbol{\Sigma}_t \hat{\boldsymbol{g}}_{ij}^t \right\|^2 \leq \frac{1}{N} \sum_{j=1}^N \|\boldsymbol{\Sigma}_t \hat{\boldsymbol{g}}_{ij}^t\|^2 \leq H^2 \|\boldsymbol{\sigma}_t\|_\infty^2 \|\boldsymbol{z}_j\|^4. \tag{121}$$

Noticed that $\mathbb{E}_{\boldsymbol{z}}[\|\boldsymbol{z}\|^4] \leq (d+4)^2$ and

$$\|\boldsymbol{\sigma}_t\|_\infty \leq \frac{1}{\|\boldsymbol{\sigma}_0^{-1}\|_{min} + 2(\sum_{k=1}^t \beta_k)\xi}, \tag{122}$$

where $\|\cdot\|_{min}$ denotes the minimum element in the input. Then we have

$$\mathbb{E}\|\boldsymbol{\Sigma}_t \hat{\boldsymbol{g}}_i^t\|_2^2 \leq H^2 \|\boldsymbol{\sigma}_t\|_\infty^2 (d+4)^2 \leq \frac{H^2(d+4)^2}{4\xi^2(\sum_{k=1}^t \beta_k)^2}, \tag{123}$$

where we reach the conclusion.

(c): We have

$$\|\hat{\boldsymbol{g}}_{ij}^t\|^2 = \|\boldsymbol{\sigma}_t^{-\frac{1}{2}} \odot \boldsymbol{z}_j (F_i(\boldsymbol{\mu}_t + \boldsymbol{\sigma}_t^{\frac{1}{2}} \odot \boldsymbol{z}_j) - F_i(\boldsymbol{\mu}_t))\|^2 \tag{124}$$

$$= \|\boldsymbol{\sigma}_t^{-\frac{1}{2}} \odot \boldsymbol{z}_j\|^2 (F_i(\boldsymbol{\mu}_t + \boldsymbol{\sigma}_t^{\frac{1}{2}} \odot \boldsymbol{z}_j) - F_i(\boldsymbol{\mu}_t))^2 \tag{125}$$

$$\leq \|\boldsymbol{\sigma}_t^{-\frac{1}{2}} \odot \boldsymbol{z}_j\|^2 H^2 \|\boldsymbol{\sigma}_t^{\frac{1}{2}} \odot \boldsymbol{z}_j\|^2 \tag{126}$$

$$\leq H^2 \|\boldsymbol{\sigma}_t^{-\frac{1}{2}}\|_\infty^2 \times \|\boldsymbol{\sigma}_t^{\frac{1}{2}}\|_\infty^2 \times \|\boldsymbol{z}_j\|^4. \tag{127}$$

Then we obtain

$$\mathbb{E}_{\boldsymbol{z}}[\|\hat{\boldsymbol{g}}_{ij}^t\|^2] \leq H^2 \|\boldsymbol{\sigma}_t^{-\frac{1}{2}}\|_\infty^2 \times \|\boldsymbol{\sigma}_t^{\frac{1}{2}}\|_\infty^2 \times \mathbb{E}[\|\boldsymbol{z}_j\|^4] \tag{128}$$

$$\leq H^2 \|\boldsymbol{\sigma}_t^{-\frac{1}{2}}\|_\infty^2 \times \|\boldsymbol{\sigma}_t^{\frac{1}{2}}\|_\infty^2 (d+4)^2. \tag{129}$$

Note that for $N$ i.i.d samples $\boldsymbol{z}_j$, we have

$$\mathbb{V}_{\boldsymbol{z}}[\frac{1}{N} \sum_{j=1}^N \hat{\boldsymbol{g}}_{ij}^t] = \frac{1}{N} \mathbb{V}_{\boldsymbol{z}}[\hat{\boldsymbol{g}}_{ij}^t] \leq \frac{1}{N} \mathbb{E}_{\boldsymbol{z}}[\|\hat{\boldsymbol{g}}_{ij}^t\|_2^2] \tag{130}$$

$$\leq \frac{H^2(d+4)^2 \|\boldsymbol{\sigma}_t^{-\frac{1}{2}}\|_\infty^2 \|\boldsymbol{\sigma}_t^{\frac{1}{2}}\|_\infty^2}{N}. \tag{131}$$

Note that we have

$$\boldsymbol{\sigma}_0^{-1} + 2b \sum_{k=1}^t \beta_t \mathbf{1} \geq \boldsymbol{\sigma}_t^{-1} \geq \boldsymbol{\sigma}_0^{-1} + 2\xi \sum_{k=1}^t \beta_t \mathbf{1}. \tag{132}$$

Then, we have

$$\|\boldsymbol{\sigma}_t^{-\frac{1}{2}}\|_\infty^2 = \|\boldsymbol{\sigma}_t^{-1}\|_\infty \leq \|\boldsymbol{\sigma}_0^{-1}\|_\infty + 2(\sum_{k=1}^t \beta_k)b. \tag{133}$$

And

$$\|\boldsymbol{\sigma}_t^{\frac{1}{2}}\|_\infty^2 = \|\boldsymbol{\sigma}_t\|_\infty \leq \frac{1}{\|\boldsymbol{\sigma}_0^{-1}\|_{min} + 2(\sum_{k=1}^t \beta_k)\xi}, \tag{134}$$

where $\|\cdot\|_{min}$ denotes the minimum element in the input.

we then have

$$\|\boldsymbol{\sigma}_t^{\frac{1}{2}}\|_\infty^2 \|\boldsymbol{\sigma}_t^{-\frac{1}{2}}\|_\infty^2 \leq \frac{\|\boldsymbol{\sigma}_0^{-1}\|_\infty + 2(\sum_{k=1}^t \beta_k)b}{\|\boldsymbol{\sigma}_0^{-1}\|_{min} + 2(\sum_{k=1}^t \beta_k)\xi} \tag{135}$$

$$= \frac{b}{\xi} + \frac{\|\boldsymbol{\sigma}_0^{-1}\|_\infty - \frac{b}{\xi}\|\boldsymbol{\sigma}_0^{-1}\|_{min}}{\|\boldsymbol{\sigma}_0^{-1}\|_{min} + 2(\sum_{k=1}^t \beta_k)\xi}. \tag{136}$$

If $\|\boldsymbol{\sigma}_0^{-1}\|_\infty - \frac{b}{\xi}\|\boldsymbol{\sigma}_0^{-1}\|_{min} \geq 0$, we have

$$\|\boldsymbol{\sigma}_t^{\frac{1}{2}}\|_\infty^2 \|\boldsymbol{\sigma}_t^{-\frac{1}{2}}\|_\infty^2 \leq \frac{b}{\xi} + \frac{\|\boldsymbol{\sigma}_0^{-1}\|_\infty - \frac{b}{\xi}\|\boldsymbol{\sigma}_0^{-1}\|_{min}}{\|\boldsymbol{\sigma}_0^{-1}\|_{min}} \leq \|\boldsymbol{\sigma}_0^{-1}\|_\infty. \tag{137}$$

If $\|\boldsymbol{\sigma}_0^{-1}\|_\infty - \frac{b}{\xi}\|\boldsymbol{\sigma}_0^{-1}\|_{min} < 0$, we have

$$\|\boldsymbol{\sigma}_t^{\frac{1}{2}}\|_\infty^2 \|\boldsymbol{\sigma}_t^{-\frac{1}{2}}\|_\infty^2 \leq \frac{b}{\xi}. \tag{138}$$

Therefore, let $C = \max(\frac{b}{\xi}, \|\boldsymbol{\sigma}_0^{-1}\|_\infty)$, we have

$$\mathbb{V}_{\boldsymbol{z}}[\|\frac{1}{N}\sum_{j=1}^N \hat{\boldsymbol{g}}_{ij}^t\|^2] \leq \frac{H^2(d+4)^2 C}{N}, \tag{139}$$

where we reach the conclusion.

### D.5    PROOF OF LEMMA B.5

We have

$$\mathbb{V}_{\boldsymbol{z}}[\boldsymbol{\lambda}^t] = \mathbb{E}_{\boldsymbol{z}}[\|\boldsymbol{\lambda}^t - \mathbb{E}_{\boldsymbol{z}}[\boldsymbol{\lambda}^t]\|^2]] \leq \mathbb{E}_{\boldsymbol{z}}[\|\boldsymbol{\lambda}^t - \boldsymbol{\lambda}^{t-1}\|^2] \tag{140}$$

$$= \mathbb{E}_{\boldsymbol{z}}[\|\gamma_t(\tilde{\boldsymbol{\lambda}}^t - \boldsymbol{\lambda}^{t-1})\|^2] \leq \gamma_t^2 \mathbb{E}_{\boldsymbol{z}}[\|\tilde{\boldsymbol{\lambda}}^t - \boldsymbol{\lambda}^{t-1}\|^2] \leq 2\gamma_t^2, \tag{141}$$

where we reach the conclusion.

### D.6    PROOF OF LEMMA B.6

According to Lemma B.4 (a) and (c), we know that $\hat{\boldsymbol{g}}_i^t$ is an unbiased estimator of the gradient $\nabla_{\boldsymbol{\mu}} J_i(\boldsymbol{\theta}_t)$ and the variance of $\hat{\boldsymbol{g}}_i^t$ is bounded. Therefore let $\boldsymbol{q}_t = -\sum_{i=1}^m \lambda_i^t \hat{\boldsymbol{g}}_i^t$, the results in Lemma B.6 can be directly obtained by Lemma 1, 7, and 8 in Zhou et al. (2022b).

## E    UPDATED RULE UNDER TRANSFORMATION

To avoid the scaling problem, we can employ monotonic transformation for the aggregated objective, i.e. $h(\boldsymbol{\lambda}^\top F(\boldsymbol{x}_j)) = \frac{\boldsymbol{\lambda}^\top F(\boldsymbol{x}_j) - \hat{\mu}}{\hat{\sigma}}$, where $\hat{\mu}$ and $\hat{\sigma}$ denote mean and stand deviation of aggregated function values $\boldsymbol{\lambda}^\top F(\boldsymbol{x}_j) = \sum_{i=1}^m \lambda_i F_i(\boldsymbol{x}_j), j = 1, \dots, N$. Then by applying this rescaling strategy, the update rule for $\boldsymbol{\mu}_t$ and $\Sigma_t$ in $t$-th iteration can be written as

$$\boldsymbol{\mu}_{t+1} = \boldsymbol{\mu}_t - \frac{\beta_t}{N}\sum_{j=1}^N (\boldsymbol{x}_j - \boldsymbol{\mu}_t)\frac{\sum_{i=1}^m \lambda_i^t F_i(\boldsymbol{x}_j) - \hat{\mu}^t}{\hat{\sigma}^t}, \tag{142}$$

$$\Sigma_{t+1}^{-1} = \Sigma_t^{-1} + \frac{\beta_t}{N}\sum_{j=1}^N \text{diag}\Big[\Sigma_t^{-1}\big[\text{diag}\big((\boldsymbol{x}_j - \boldsymbol{\mu}_t)(\boldsymbol{x}_j - \boldsymbol{\mu}_t)^\top \Sigma_t^{-1}\big)\frac{\sum_{i=1}^m \lambda_i^t F_i(\boldsymbol{x}_j) - \hat{\mu}^t}{\hat{\sigma}^t}\big]\Big].$$
$$\tag{143}$$

# F    ADDITIONAL MATERIALS FOR SECTION 6

## F.1    SYNTHETIC PROBLEMS

**Evaluation Metrics.**    The Pareto optimal set of problem (19) is $\mathcal{P}_1 = \{\boldsymbol{x} \mid x_i \in [-0.01, 0.01]\}$. Therefore. the result is evaluated by calculating the Euclidean distance between solution $\boldsymbol{x}$ and the set $\mathcal{P}_1$, which is denoted $\mathcal{E} = \text{dist}(\boldsymbol{x}, \mathcal{P}_1)$. We denote the Pareto optimal set of problem (20) as $\mathcal{P}_2$. Since the Pareto front of problem (20) is concave, the solution of the ASMG method will go to the boundary of its Pareto optimal set, i.e. $\hat{\mathcal{P}}_2 = \{\boldsymbol{x} \mid x_i \in \{-0.1, 0.1\}\} \subset \mathcal{P}_2$. Moreover, For ES and CMA-ES methods, since their optimization objective is $F_1(x) + F_2(x)$, the corresponding solution set is $\hat{\mathcal{P}}_2$. Therefore, the result of these three methods on problem (20) is evaluated by $\mathcal{E} = \text{dist}(\boldsymbol{x}, \hat{\mathcal{P}}_2)$. The Pareto optimal set of problem (21) is $\mathcal{P}_3 = \{\boldsymbol{x} \mid \boldsymbol{x} = 0\}$. Therefore, the result is evaluated by $\mathcal{E} = \text{dist}(\boldsymbol{x}, \mathcal{P}_3)$.

**Implementation Details.**    For all the methods, we initialize $\boldsymbol{\mu}_0$ from the uniform distribution $\text{Uni}[0, 1]$, and set $\boldsymbol{\Sigma}_0 = \boldsymbol{I}$. The ASMG method uses a fixed step size of $\beta = 0.1$ and $\gamma_t = 1/(t+1)$. For the ES method, we employ the default step size from Salimans et al. (2017), i.e., $\beta = 0.01$. For the BES method, we adopt the default step size from Gao & Sener (2022), i.e., $\beta = 0.01$. We employ the default hyperparameter setting from He et al. (2020) for the MMES method. We then assess these methods using varying sample sizes, i.e. $N \in \{10, 50, 100\}$. The mean value of $\mathcal{E}$ over 3 independent runs is reported.

**Result.**    Figure 2 and 3 show the results on three 100-dimensional synthetic problems with sample sizes $N = 10$ and $N = 100$, respectively. Combining these results with the result from Figure 1, we observe consistent performance from the proposed ASMG method, consistently achieving high precision across all three cases, i.e. $10^{-4}$, for $N = 10, 50, 100$. The CMA-ES method shows convergence with high precision on the Shift $l_1$-Ellipsoid problem when $N = 10$ and $50$. However, it fails to converge when the sample size is very small, i.e., $N = 10$. The same performance also occurs on the Shift $l_{\frac{1}{2}}$-Ellipsoid problem for the CMA-ES method. It can achieve $10^{-1}$ precision when $N = 50$ and $100$, but only achieve $10^1$ precision when $N = 10$. It still fails on the Mixed Ellipsoid-Rastrigin10 problem when $N = 50$ and $100$. The MMES method also cannot reach a high precision on these problems. The ES and BES methods do not converge in any of the settings, indicating that it could be challenging for these methods to optimize these non-smooth or non-convex problems. These results show the effectiveness of the proposed ASMG method.

Figure 4 presents the results for the shift $l_{\frac{1}{2}}$-ellipsoid problem with a sample size of $N = 100$ across various problem dimensions, i.e. $d \in \{200, 500, 1000\}$. The CMA-ES method can still converge when $d = 200$, but it does not converge when $d = 1000$. In contrast, the ASMG method consistently achieves high precision across all three settings, demonstrating its effectiveness in handling high-dimensional problems.

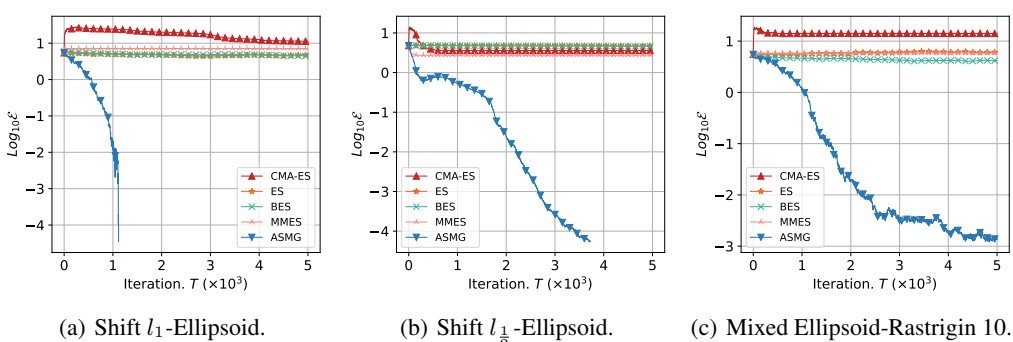

(a) Shift $l_1$-Ellipsoid.            (b) Shift $l_{\frac{1}{2}}$-Ellipsoid.            (c) Mixed Ellipsoid-Rastrigin 10.

Figure 2: Results on the synthetic problems with 10 samples (i.e., $N = 10$).

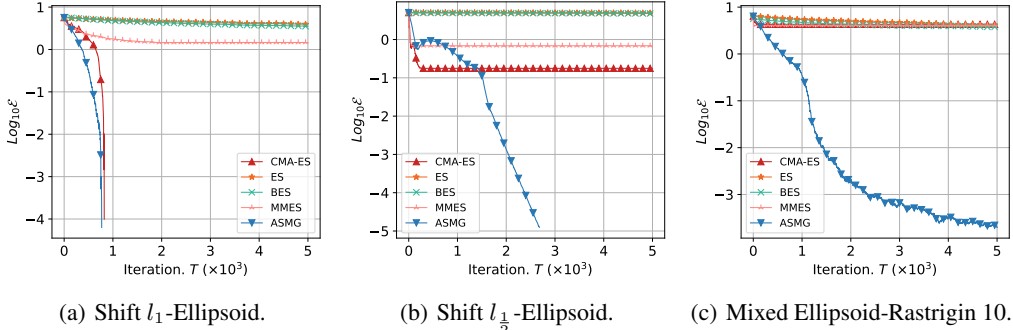

(a) Shift $l_1$-Ellipsoid.  (b) Shift $l_{\frac{1}{2}}$-Ellipsoid.  (c) Mixed Ellipsoid-Rastrigin 10.

Figure 3: Results on the synthetic problems with 100 samples (i.e., $N = 100$).

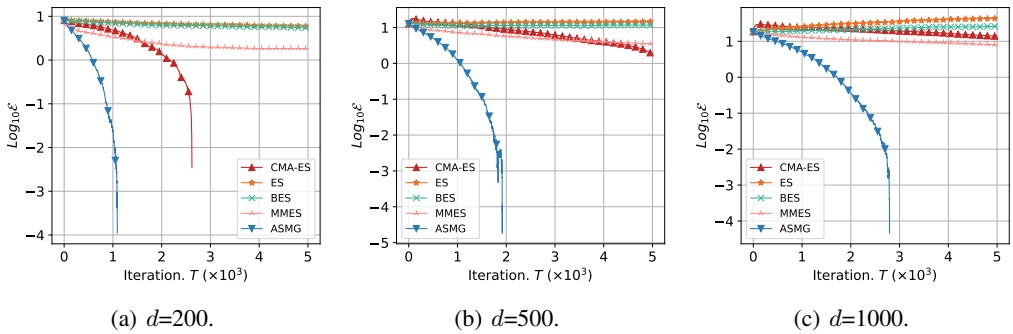

(a) $d$=200.  (b) $d$=500.  (c) $d$=1000.

Figure 4: Results on the shift $l_1$-ellipsoid problem with $N = 100$ and different problem dimension $d$.

### F.2 BLACK-BOX MULTI-TASK LEARNING

**Details of CLIP.** CLIP is a widely adopted vision-language model that trains an image encoder $h_{\text{image}}(\cdot)$ and a text encoder $h_{\text{text}}(\cdot)$ jointly by aligning the embedding space of images and text. Given an image $x$ and a set of class names $\{y_i\}_{i=1}^K$, CLIP obtain image features $h_{\text{image}}(x)$ and a set of text features $\{h_{\text{text}}(p; y_i)\}_{i=1}^K$ where $p \in \mathbb{R}^D$ represents the token embedding of the shared prompt. The image $x$ is classified into the class $y_i$ that corresponds to the highest similarity score $h_{\text{image}}(x) \cdot h_{\text{text}}(p; y_i)$ among the cosine similarities between the image features and all the text features. In the zero-shot setup, the shared token embedding $p$ is transformed from the prompt "a photo of a", while in the prompt tuning setup, the token embedding of the shared prompt is optimized directly to enhance performance.

**Loss function $\mathcal{L}_i$.** In the context of multi-task learning, we consider a scenario involving $m$ tasks, each having its own dedicated training dataset. For task $i$, we have dataset $\mathcal{D}_i = \{(x_k, \hat{y}_k)\}$. For each training epoch, we sample a mini-batch $\mathcal{B}_i$ from $\mathcal{D}_i$ and the function $\mathcal{L}_i$ in Eq. (22) can be formulated as

$$\mathcal{L}_i(B_i; \{\mathcal{M}_c, Av\}) = \sum_{(x, \bar{y}) \in \mathcal{B}_i} \ell(\mathcal{M}_c(Av; x), \bar{y}),$$

where $\ell$ can be cross-entropy function for classification problem and $\mathcal{M}_c$ denotes CLIP model in our setting.

**Datasets.** We conduct experiments on two MTL benchmark datasets (Lin & Zhang, 2023), i.e., *Office-31* (Saenko et al., 2010) and *Office-home* (Venkateswara et al., 2017). The *Office-31* dataset includes images from three different sources: Amazon (**A**), digital SLR cameras (**D**), and Webcam (**W**). It contains 31 categories for each source and a total of 4652 labeled images. The *Office-home* dataset includes images from four sources: artistic images (**Ar**), clip art (**Cl**), product images (**Pr**), and real-world images (**Rw**). It contains 65 categories for each source and a total of 15,500 labeled images. For those two datasets, we treat the multi-class classification problem on each source as a separate task.

**Implementation Details.** Following the setup of Zhou et al. (2022a), we conduct experiments based on the CLIP model with ResNet-50 as the image encoder and use a prompt with 4 tokens for the text encoder where both the image and text encoders are kept frozen during the experiments. For zero-shot, we apply the default prompt "a photo of a {class}".

For all methods, we set $\boldsymbol{\mu}_0 = \mathbf{0}$ and $\boldsymbol{\Sigma}_0 = \boldsymbol{I}$ as initialization. For all baseline methods except the zero-shot setting, we optimize the prompt with a batch size of 64 for 200 epochs, the population size $N$ is set as 20 for *Office-31* and 40 for *Office-home*, while $\boldsymbol{A}$ is sampled from the normal distribution as described in the Sun et al. (2022a), i.e. $\mathcal{N}(0, \frac{\sigma_e}{\sqrt{d}})$, where $\sigma_e$ is the standard deviation of word embeddings in CLIP. For ASMG and ASMG-EW methods, the step size is fixed as $\beta = 0.5$. The coefficient $\gamma$ in the ASMG method is set as $\gamma_t = 1/(t+1)$. For the ES method, we employ the default step-size of ES in Salimans et al. (2017), i.e., 0.01. For the BES method, we perform grid search on step size, i.e., $\beta$ is chosen from $\{0.5, 0.1, 0.01\}$. For the MMES method, we employ the default hyperparameter setting from He et al. (2020). Additionally, we evaluate the performance of the method on different dimensions of $\boldsymbol{z}$, specifically $d \in \{256, 512, 1024\}$. The CMA-ES method is implemented using the official implementation available [1] while we implement the ES and BES method by ourselves.

## G    RELATIONSHIP TO GRADIENT-BASED MOO

For previous methods on MOO, the most relevant method to our approach is gradient-based MOO methods (Yu et al., 2020; Liu et al., 2021; Fernando et al., 2022; Zhou et al., 2022b), as they also aim at finding the Pareto optimal solution or the Pareto stationary solution. A typical gradient-based method is the MGDA method (Désidéri, 2012), which also solves a max-min optimization problem to obtain the weights. However, the proposed ASMG method is not a typical MGDA-type method. The max-min optimization problem proposed in MGDA-type methods is related to the true gradient of the parameters. They add a regularization term $\|d\|^2$ to control the norm of the aggregated gradient. In our case, the update is conducted on a Gaussian distribution, and we need to jointly update the mean and covariance matrix. We use Kullback-Leibler divergence to regularize the distance between two distributions, i.e. $\theta$ and $\theta_t$. Therefore, the form of the proposed max-min optimization, i.e. Eq. (4), differs from MGDA-type methods, and the solution process is also different. However, our max-min problem can also lead to a simple quadratic programming problem for the aggregation weights computation.

---

[1] https://github.com/CMA-ES/pycma

