# OpenReview forum: "Adaptive Stochastic Gradient Algorithm for Black-box Multi-Objective Learning"
_ICLR.cc/2024/Conference — ICLR 2024 poster_

### Official Review · Reviewer_w58H · 2023-10-31

**Soundness:** 3 good
**Presentation:** 3 good
**Contribution:** 3 good
**Rating:** 6
**Confidence:** 3

**Summary:**

In this paper, the authors propose a gradient-free black-box multi-objective optimization (MOO) method with dynamic weighting of objectives, leveraging concepts developed in the gradient based multi-objective optimization literature. The proposed method achieves this by deriving a gradient-free estimates for the objective gradients with respect to the parameters of the search distribution, and solving for dynamic weights for each objective calculated based on the aforementioned estimates. The authors provide relationship between the solutions to the original MOO problem and the solutions of the Gaussian smoothed MOO problem, which the proposed method find. The authors further provide theoretical convergence guarantees for the proposed method, and provide empirical validation for the validity of the proposed method.

**Strengths:**

* The paper provides a black-box multi-objective optimization algorithm in the stochastic setting with theoretical convergence guarantees, which is a novel contribution to my knowledge.

* The paper provide empirical evidence for the efficacy of the proposed method over existing methods using synthetic and real world benchmarks.

* The paper is fairly easy to read, and the ideas are presented well.

**Weaknesses:**

* The proposed method assumes full access to the objective function, while in most applications (like in multi-task learning setup used in the experiment setup), full evaluation of objectives might not be possible or might be costly. The theoretical guarantees provided does not seem to cover this. However, the authors provide empirical evidence for the proposed method in this setting.

* The setting considered might be not suitable for very large parameter spaces (i.e. when d is very large), which is the case when the decision variable is represented by a deep neural network.

Minor comments:

* The Pareto stationarity defined in Proposition 2.3. may not be a necessary condition for Pareto optimality when $\mathcal{X} \subset \mathbb{R}^d$ (Consider when there is a global minimum for all the objectives at a boundary of  $\mathcal{X}$)

**Questions:**

* How doe the proposed method scale with the size of the parameter to be learned?

* How sensitive is the proposed method to the stochasticity of the objectives (i.e. when the average over all data-points is estimated by the average of over a mini batch)?

---

> ### Author Response · Authors · 2023-11-18
> **Reply to Reviewer w58H**
>
> Thank you for your thoughtful review and constructive comments.
>
> ---
>
> >Q1. "The proposed method assumes full access to the objective function, while in most applications (like in multi-task learning setup used in the experiment setup), full evaluation of objectives might not be possible or might be costly. The theoretical guarantees provided does not seem to cover this. However, the authors provide empirical evidence for the proposed method in this setting.''
>
> **A1**: It is challenging to obtain the full evaluation of the objective for many practice problems. Therefore, we can only approximate the loss function with a min-batch of data in our multi-task learning setup. It is interesting to analyze the effect of the stochasticity caused by mini-batch approximation to our algorithm theoretically, and we regarded it as a future work.
>
> ---
>
> >Q2. "The setting considered might be not suitable for very large parameter spaces (i.e. when d is very large), which is the case when the decision variable is represented by a deep neural network.''
>
> **A2**: The black-box optimization method can not deal with extremely large parameter spaces, such as optimizing the parameters of large deep neural network models. One reason is the difficulty in approximating a suitable search direction for a highly-dimensional parameter with limited function queries. However, the studied setting of black-box multi-task prompt learning is still very important, especially with the emergence of large language models and many popular APIs that are only accessible for inference. Recently, the use of black-box optimization methods for prompt learning in large language models has gained considerable attention [1,2], demonstrating promising applications in various domains.
>
> ---
>
> >Q3. "The Pareto stationarity defined in Proposition 2.3. may not be a necessary condition for Pareto optimality when ...''
>
> **A3**: Thanks for your insightful comments. In the general nonconvex cases, our method focuses on finding the Pareto stationary solution. We have corrected the mentioned claim about Pareto stationarity in Section 2.1 of the updated version.
>
> ---
>
> >Q4. "How doe the proposed method scale with the size of the parameter to be learned?''
>
> **A4**: We now provide the scalability analysis of our algorithm. For the proposed ASMG method, the computational cost of solving the weights in each iteration is negligible since the number of objectives is usually very small (i.e. $d>>m$). Therefore, the computational cost in each iteration is of order $\mathcal{O}(mNd)$. We provide an ablation study of our method on different dimensions, i.e. $\{d=200,500,1000\}$, and the results (Figure 7) are put in Appendix F. The results demonstrate the effectiveness of the proposed ASMG method on high-dimensional problems.
>
> ---
>
> >Q5. "How sensitive is the proposed method to the stochasticity of the objectives (i.e. when the average over all data-points is estimated by the average of over a mini batch)?"
>
> **A5**: We conduct an experiment of the proposed method with different batch sizes on the Office-31 dataset with $d=512$. The average classification accuracy with different batch sizes is reported in the following table. The result shows that the proposed method is not very sensitive to the stochasticity of the objectives. We will complete this table in the revision.
>
> \begin{array}{ccccc}
>     \hline
>     \text{Batch Size} & 32 & 64 & 128 &  256 \newline
>     \hline
>     \text{ASMG} & 81.78 & 81.47 & 81.96 & 81.48\newline
>     \hline
> \end{array}
>
> ---
>
> **References**
>
> [1] Sun et al. Black-box tuning for language-model-as-a-service. ICML, 2022.
>
> [2] Sun et al. Multitask pre-training of modular prompt for chinese few-shot learning. ACL, 2023.

---

> ### Author Response · Authors · 2023-11-21
> **We hope that our responses addressed your concerns**
>
> Dear reviewer w58H,
>
> Thank you again for taking the time to review our paper, and giving us insightful comments to improve the paper. We are pleased that the reviewer found our proposed method to be a **novel contribution**.  We hope our responses adequately address your concerns. If you have any additional questions, we would be happy to follow up.
>
> With gratitude,
>
> Authors

---

### Official Review · Reviewer_ZfhH · 2023-11-01

**Soundness:** 2 fair
**Presentation:** 2 fair
**Contribution:** 3 good
**Rating:** 6
**Confidence:** 3

**Summary:**

This paper proposes a MGDA-type gradient-based algorithm for solving the problem of black-box multi-objective optimization. Inspired by the stochastic gradient approximation method in black-box optimization, the objective of MOO is replaced by the expected loss where the decision is sampled under some parametric search distribution. Then the MOO problem is transformed into the problem of finding the optimal search distribution with minimal expected loss. When assuming the search distribution to be Gaussian, the gradient of the distribution parameters (mean and variance) is accessible, making it possible to apply gradient-based MOO algorithms. The relation between the two problems and the convergence rate are analyzed. Numerical experiments are conducted to verify the effectiveness of the proposed algorithm compared to evolutionary algorithms.

**Strengths:**

1. The idea of applying stochastic gradient approximation method to black-box MOO problem sounds novel.

2. This paper is technically sound. The analysis on the relation between the original black-box MOO problem and the smoothed version in convex/non-convex cases is interesting.

3. This paper is well written in general and easy to follow.

**Weaknesses:**

1. The related work section only includes evolutionary methods and BO for multi-objective black-box optimization. The relevance to previous works on MOO/stochastic gradient approximation should also be clarified. A detailed discussion of technical difficulties in applying stochastic gradient approximation to black-box MOO may help better highlight the significance and technical novelty of this paper.

2. The experiment considers evolutionary algorithm as the only baseline, which seems insufficient. There are also other major branches of black-box MOO algorithms, e.g., multi-objective Bayesian optimization [Zhang & Golovin, 2020] or multi-objective bandits, that should be compared.

**Questions:**

1. The proposed method assumes that the covariance matrix of the search distribution is a diagonal matrix. As I understand, this assumption indicates that the different dimensions of x are independent. Can you justify this assumption? I wonder if there is any performance gap with/without using this assumption.

2. The Monte-Carlo sampling approach for calculating the expectation may bring additional computational complexity to the proposed algorithm. Can you analyze the complexity or provide some empirical results on the real running time?

---

> ### Author Response · Authors · 2023-11-18
> **Reply to Reviewer ZfhH (1/3)**
>
> Thank you for your thoughtful review and constructive comments.
>
> ---
>
> >Q1. "The related work section only includes evolutionary methods and BO for multi-objective black-box optimization. The relevance to previous works on MOO/stochastic gradient approximation should also be clarified. A detailed discussion of technical difficulties in applying stochastic gradient approximation to black-box MOO may help better highlight the significance and technical novelty of this paper.''
>
> **A1**:  We provide a discussion about the previous works on MOO and stochastic gradient approximation.
>
> (i) For previous methods on MOO, the most relevant method to our approach is gradient-based MOO methods, as they also aim at finding the Pareto optimal solution or the Pareto stationary solution. A typical gradient-based method is the MGDA method, which also solves a max-min optimization problem to obtain the weights. However, **the proposed ASMG method is not a typical MGDA-type method**. The max-min optimization problem proposed in MGDA-type methods is related to the true gradient of the parameters. They add a regularization term $||d||^2$ to control the norm of the aggregated gradient. In our case, the update is conducted on a Gaussian distribution, and we need to jointly update the mean and covariance matrix. We use Kullback–Leibler divergence to regularize the distance between two distributions, i.e. $\theta$ and $\theta_t$. Therefore, the form of proposed max-min optimization (Eq. (4)) is **completely different** from MGDA-type methods (e.g. Eq. (2) in [4]), and the solution process is also different (Appendix A.1). Moreover, it is interesting that our novel max-min problem leads to a simple quadratic programming problem for the aggregation weights computation.
>
>
> (ii) The prior studies on stochastic gradient approximation have not specifically addressed the black-box MOO problem. Directly applying weighted sum aggregation of multi-objectives to a single objective is not effective, as observed in our experiments. To address this challenge, we propose an adaptive aggregation by introducing a novel max-min optimization problem (i.e. Eq. (4)).
>
>
> The main technical difficulty of applying stochastic gradient approximation to black-box MOO is to propose an effective optimization method to update the distribution with a convergence guarantee and verify the performance in different experimental settings. The algorithm proposed in our work successfully resolves these technical difficulties, and possesses significant technical novelty in three key aspects:
>
> (i) we introduce a novel max-min optimization problem (i.e. Eq. (4)) for the Gaussian-smoothed objectives to solve black-box MOO. This approach is crucial as previous MGDA-based methods are not applicable in the absence of the true gradients. The max-min optimization problem provides update rules for both the mean and covariance matrix, and notably, we uncover that it can derive weights as a solution to a quadratic programming problem;
>
> (ii) we explore the connection of the optimal conditions between the original MOO and the corresponding Gaussian-smoothed MOO, and we prove the convergence rate for the proposed algorithm in both convex and non-convex scenarios. These original findings provide a theoretical guarantee for our algorithm;
>
> (iii) we are the first to study such a black-box multi-task learning problem (i.e. Eq. (22)) as a black-box MOO. Our experiment on the black-box multi-task learning setting based on the CLIP model demonstrates the effectiveness of the proposed ASMG method in real-world applications.
>
> Thanks for your valuable comments. We added these discussions in Section G of the updated version.
>
> ---

---

> ### Author Response · Authors · 2023-11-18
> **Reply to Reviewer ZfhH (2/3)**
>
> >Q2. "The experiment considers evolutionary algorithm as the only baseline, which seems insufficient. There are also other major branches of black-box MOO algorithms, e.g., multi-objective Bayesian optimization [Zhang \& Golovin, 2020] or multi-objective bandits, that should be compared.''
>
> **A2**: One motivation behind this paper is to develop a method applicable to some real-world black-box MOO scenarios, such as black-box multi-task prompt tuning. In such scenarios, the problem's dimension is relatively high, and most Bayesian optimization methods cannot handle it. The multi-objective Bayesian optimization [1] conducts experiments on the problems up to 24 dimensions. In contrast, our algorithm can handle high dimensional back-box MOO problems ($d\ge 1000$) on both synthetic and black-box multi-task learning (MTL) settings (Figure 7 and Table 1). Besides, the code of [1] has not been released and we did not find an official implementation of multi-objective bandits, making it challenging to include those baselines in our comparisons.
>
> To enrich our experiments, we conducted additional experiments on two recently proposed methods: BES, a random search method [2], and MMES, an evolution strategy method [3]. The results of these two methods on both synthetic and black-box multi-task learning problems are put in Appendix F of the updated version. The results consistently demonstrate the effectiveness of the proposed ASMG method. The following tables show part of the result of these two baseline methods on the black-box multi-task learning settings. The first and the second tables show the experimental results on Office-31 and Office-home datasets, respectively.
>
> \begin{array}{ccccc}
>     \hline
>     \text{Method} & \text{A} & \text{D} & \text{W} & \text{Avg} \newline
>     \hline
>     \text{BES (d=256)} &  72.65 & 80.60 & 73.52 & 75.59\newline
>     \text{MMES (d=256)} & 75.90 & 83.33 & 76.67 & 78.63\newline
>     \text{ASMG (d=256)} & 77.83 & 86.61 & 80.56 & \textbf{81.67}\newline
>     \hline
>     \text{BES (d=512)} & 75.73 & 82.51 & 74.81 & 77.68 \newline
>     \text{MMES (d=512)} & 76.01 & 84.70 & 77.22 & 79.31\newline
>     \text{ASMG (d=512)} & 78.63 & 87.43 & 78.33 & \textbf{81.47}\newline
>     \hline
>     \text{BES (d=1024)} & 72.14 & 79.51 & 71.67 & 74.44\newline
>     \text{MMES (d=1024)} & 77.09 & 81.42 & 75.74 & 78.09\newline
>     \text{ASMG (d=1024)} & 76.30 & 87.70 & 80.19 & \textbf{81.40}\newline
>     \hline
> \end{array}
>
> \begin{array}{cccccc}
>     \hline
>     \text{Method} & \text{Ar} & \text{Cl} & \text{Pr} & \text{Rw} & \text{Avg} \newline
>     \hline
>     \text{BES (d=256)} & 68.94 & 45.97 & 81.53 & 79.42 & 68.97\newline
>     \text{MMES (d=256)} & 71.85 & 49.26 & 82.10 & 81.37 & 71.14\newline
>     \text{ASMG (d=256)} & 74.26 & 53.52 & 86.23 & 83.03 & \textbf{74.26}\newline
>     \hline
>     \text{BES (d=512)} & 69.39 & 48.18 & 82.94 & 80.29 & 70.20\newline
>     \text{MMES (d=512)} & 70.40 & 50.45 & 85.10 & 82.56 & 72.13\newline
>     \text{ASMG (d=512)} & 73.50 & 52.84 & 85.88 & 83.78 & \textbf{74.00}\newline
>     \hline
>     \text{BES (d=1024)} & 70.27 & 48.25 & 79.94 & 80.00 & 69.62\newline
>     \text{MMES (d=1024)} & 71.03 & 49.19 & 84.29 & 81.95 & 71.61\newline
>     \text{ASMG (d=1024)} & 73.18 & 51.82 & 85.84 & 83.21 & \textbf{73.51}\newline
>     \hline
> \end{array}
>
> ---

---

> ### Author Response · Authors · 2023-11-18
> **Reply to Reviewer ZfhH (3/3)**
>
> >Q3. "The proposed method assumes that the covariance matrix of the search distribution is a diagonal matrix. As I understand, this assumption indicates that the different dimensions of x are independent. Can you justify this assumption? I wonder if there is any performance gap with/without using this assumption.''
>
> **A3**: We employ the diagonal covariance update scheme to approximate the full covariance matrix update for fast computation. The diagonal update is best suited for separable functions (different dimensions of x are independent). For non-separable functions, it may not be as good as a full-covariance update scheme. However, the full-covariance update scheme needs to compute the inverse of the covariance matrix, which needs $O(d^3)$.  In practical tasks, e.g., prompt fine-tuning tasks,  the dimension of tasks is generally large. In our experiments, $d=\{256, 512,1024\}$. In such cases, the $O(d^3)$ computation cost is too expensive and prohibitable. Thus, the diagonal covariance update scheme with a $O(d)$ update instead of $O(d^3)$ is necessary.
>
> In our experiments on practical prompt fine-tuning tasks, the underlying black-box functions generally are non-separable (each dimension of x is not independent).  Empirical results on these non-separable tasks demonstrate the performance improvement of our method compared with other baselines, including CMA-ES, BES [2], and MMES [3],  which shows the capabilities of our methods in handling the non-separable tasks.
>
> ---
>
> >Q4. "The Monte-Carlo sampling approach for calculating the expectation may bring additional computational complexity to the proposed algorithm. Can you analyze the complexity or provide some empirical results on the real running time?''
>
> **A4**: Since the ES-type approaches, e.g., CMAES, ES,  MMES[3], all take Monte-Carlo sampling in the parameter update. Our method does not incur additional time complexity costs. For our method, the computational cost of solving the weights in each iteration is negligible since the number of objectives is usually very small (i.e. $d>>m$). Therefore, the computational cost in each iteration is of order $\mathcal{O}(mNd)$. We report the running time (s) per iteration with various sample sizes $N$ in the black-box MTL settings with $d=512$ in the following two tables. The first and the second tables show the experimental results on Office-31 ($m=3$) and Office-home ($m=4$) datasets, respectively. All the methods achieve similar running times and the experimental results are consistent with our analysis. We added the discussion about computational complexity in Section 3.2 of the updated version.
>
> \begin{array}{cccc}
>     \hline
>     \text{Method} & N=20 & N=40 & N=80 \newline
>     \hline
>     \text{ES} & 5.110 & 9.620 & 18.61\newline
>     \text{CMA-ES} & 5.236 & 9.680 & 18.75\newline
>     \text{ASMG} & 5.121 & 9.406 & 17.96\newline
>     \hline
> \end{array}
>
> \begin{array}{cccc}
>     \hline
>     \text{Method} & N=20 & N=40 & N=80 \newline
>     \hline
>     \text{ES} & 7.473 &  14.80 & 29.48\newline
>     \text{CMA-ES} & 7.647 & 14.60 & 28.88\newline
>     \text{ASMG} & 7.553 & 14.56 & 28.67\newline
>     \hline
> \end{array}
>
> ---
>
> **References**
>
> [1] Zhang et al. Random hypervolume scalarizations for provable multi-objective
> black box optimization. ICML, 2020.
>
> [2] Gao et al. Generalizing Gaussian Smoothing for Random Search. ICML, 2022.
>
> [3] He et al. MMES: Mixture Model-Based Evolution Strategy for Large-Scale Optimization. TEVC, 2021
>
> [4] Fernando et al. Mitigating Gradient Bias in Multi-objective Learning: A Provably Convergent Stochastic Approach. ICLR, 2022.

---

> ### Author Response · Authors · 2023-11-21
> **We hope that our responses addressed your concerns**
>
> Dear reviewer ZfhH,
>
> We want to thank you again for taking the time to review our paper and giving us constructive feedback to improve the paper. As the end of the rebuttal phase approaches, we sincerely look forward to your feedback.
>
> We would highly appreciate it if you could once again help review our responses and let us know whether your concerns have been addressed. If you have any additional questions, we would be happy to follow up.
>
> Thank you in advance,
>
> Authors

---

> ### Comment · Reviewer_ZfhH · 2023-12-03
>
> Thank you for the detailed clarifications and additional experiments.

---

### Official Review · Reviewer_J6Rt · 2023-11-01

**Soundness:** 3 good
**Presentation:** 3 good
**Contribution:** 3 good
**Rating:** 6
**Confidence:** 4

**Summary:**

This paper proposes a ASMG algorithm for black-box multi-objective optimization. It is the first to design a stochastic gradient algorithm for black-box MOO with a theoretical convergence guarantee. The authors also prove the convergence rate for the proposed ASMG algorithm in both convex and non-convex cases. Empirically, the proposed ASMG algorithm achieves competitive performances on both toy examples and CLIP models.

**Strengths:**

I think the studied topic of black-box MOO problem is very important, especially with the emergence of  large language models, multi-modality models. This is a good timing to investigate on this topic given that the most popular APIs we use (ChatGPT, etc.) are only accessible for inference.

To the best of my knowledge, this paper is the first one that designs a stochastic gradient algorithm for black-box MOO. Moreover, it comes up with a theoretical convergence guarantee. For convex case, it possesses a convergence rate $\mathcal{O}\left(\frac{\log T}{T}\right)$. For non-convex case, it has a $\mathcal{O}\left(T^{-\frac{1}{2}}\right)$ convergence rate to reach a Pareto stationary solution.

I also think the CLIP example is very important. This shows that the authors do not only want to test the proposed algorithms on toy examples. They actually would like to use it on those state of art algorithms, which I really appreciate it.

**Weaknesses:**

think this paper can be improved in several ways:

1. More discussions about more related works, such as recent KDD papers about one important application of multi-objective optimization in learning to rank https://dl.acm.org/doi/abs/10.1145/3580305.3599870 and https://dl.acm.org/doi/abs/10.1145/3580305.3599482

One more example is the lack of discussion about the seminal work in MOO for Multi-task learning https://arxiv.org/pdf/1810.04650.pdf

2. How likely does the assumption that the covariance matrix Σ is a diagonal matrix hold?

**Questions:**

1. How likely does the assumption that the covariance matrix Σ is a diagonal matrix hold?


2. I'd like to see how likely the assumptions in Section 4, especially the convex case, hold? For example, in 6.1 SYNTHETIC PROBLEMS, the

---

> ### Author Response · Authors · 2023-11-18
> **Reply to Reviewer J6Rt**
>
> Thank you for your thoughtful review and constructive comments.
>
> ---
>
> >Q1. "More discussions about more related works, such as recent KDD papers about one important application of multi-objective optimization in learning to rank ...''
>
> **A1**: Thank you for your suggestion, we've added the discussions about the two papers about the learn-to-rank (LTR) problem in the introduction section of the updated version. These works propose and evaluate various gradient-based MOO methods for solving the LTR problem. Different from those weighting approaches, our method is designed for black-box MOO and we obtain the weights by solving a max-min optimization problem for the Gaussian-smoothed MOO.
>
> ---
>
> >Q2. "One more example is the lack of discussion about the seminal work in MOO for Multi-task learning [1]''
>
> **A2**: We've added the discussions about [1] in the updated version. [1] is a seminal work that successfully applies the MGDA method to multi-task learning. However, [1] and various MGDA-based methods are inapplicable to black-box multi-task learning, and our method can handle such settings.
>
> ---
>
> >Q3. "How likely does the assumption that the covariance matrix $\Sigma$ is a diagonal matrix hold?"
>
> **A3**: In our algorithm, we employ the diagonal covariance update scheme to approximate the full covariance matrix, because we need to calculate the inversion of the covariance matrix to obtain the stochastic gradients and the computational cost of calculating the inversion of a non-diagonal $d$-dimensional matrix is very high (i.e. $\mathcal{O}(d^3)$). Therefore, due to the consideration of computational cost, we assume that the covariance matrix $\Sigma$ is a diagonal matrix. Then for the proposed method, the computational cost per iteration is of order $\mathcal{O}(mNd)$, which is much more affordable. In practice, We intercept the diagonal elements of the estimated gradient of $\Sigma$ to keep the covariance matrix a diagonal matrix.
>
> ---
>
> >Q4. "I'd like to see how likely the assumptions in Section 4, especially the convex case, hold? For example, in 6.1 SYNTHETIC PROBLEMS, the"
>
> **A4**: According to our assumptions in Theorem 4.4, the objective functions should be convex and are not necessary to be differentiable, and the Gaussian-smooth objective is $c$-strongly convex.  For the convex synthetic problem, such as $l_2$-norm problem, where each objective takes form of $||\boldsymbol{x}-C\||_2^2$ and $C$ is a constant, this assumption holds. The boundedness assumption of $||\mu_t-\mu^*||$ in Theorem 4.4 can be held when we conduct our algorithm in a bounded region.
>
> ---
>
> **References**
>
> [1] Sener et al. Multi-Task Learning as Multi-Objective Optimization. NeurIPS 2018.

---

> ### Author Response · Authors · 2023-11-21
> **We hope that our responses addressed your concerns**
>
> Dear reviewer J6Rt,
>
> Thank you again for your feedback and for recognizing our work **to address an important problem with novel technical contributions**. We hope our responses adequately address your concerns. If you have any additional questions, we would be happy to follow up.
>
> Best wishes,
>
> Authors

---

> > ### Comment · Reviewer_J6Rt · 2023-11-23
> >
> > thank you very much for your response and updating the paper accordingly. I believe that the studied problem is very important.

---

> > > ### Author Response · Authors · 2023-11-23
> > >
> > > Thanks for your reply. We are glad that our reply addressed your concerns.

---

### Official Review · Reviewer_hUW7 · 2023-11-05

**Soundness:** 3 good
**Presentation:** 2 fair
**Contribution:** 3 good
**Rating:** 6
**Confidence:** 3

**Summary:**

The paper proposes a novel algorithm called Adaptive Stochastic Gradient Algorithm for Black-Box Multi-Objective Learning (ASMG) to optimize multiple potentially conflicting objectives with function queries only. The authors provide the detailed discussion of the original MOO and the corresponding Gaussian-smoothed MOO and provide theoretical proofs for the convergence rate of the algorithm. The authors also demonstrate the effectiveness of the ASMG algorithm on various numerical benchmark problems.

**Strengths:**

1. Originality: The authors provide a novel approach to black-box multi-objective optimization with the new concept of "adaptive" stochastic gradient approximation that has a theoretical convergence guarantee.

2. Quality: The paper provides a rigorous theoretical analysis of the proposed algorithm, such as proofs of convergence rate. The authors also demonstrate the effectiveness of the algorithm on multiple numerical benchmark problems, showing that it outperforms the ES method in terms of convergence speed and solution quality.

3. Clarity: The paper is well-written and easy to follow, with clear explanations of the algorithm and its theoretical properties.

4. Significance: The proposed method has the ability to optimize multiple objectives simultaneously with function queries only, which is the scenario of many MOO-based learning problems, such as tasks involving large language models that are only allowed for access with APIs.

**Weaknesses:**

1. In the introduction, lack of discussion on advantages of the Gaussian-smoothed MOO, compared with the traditional method.
2. Lack of comparison with more recent state-of-the-art methods. The authors only compare their algorithm with the ES-based method.
3. Limited discussion of the algorithm's computational complexity and runtime.

**Questions:**

1. How does the proposed algorithm compare to other black-box multi-objective optimization methods? Since only comparison with the ES-based method is discussed in the paper.
2. What is the computational complexity and runtime of the proposed algorithm compared with other black-box multi-objective optimization methods?

---

> ### Author Response · Authors · 2023-11-18
> **Reply to Reviewer hUW7 (1/2)**
>
> Thank you for your thoughtful review and constructive comments.
>
> ---
>
> >Q1. "In the introduction, lack of discussion on advantages of the Gaussian-smoothed MOO, compared with the traditional method.''
>
> **A1**: The advantage of using the Gaussian-smoothed MOO method is that we can effectively estimate stochastic gradients, and enable a more accurate search direction for the distribution. This capability is valuable in solving high-dimensional black-box MOO problems and providing comprehensive theoretical analysis.  In contrast, the Bayesian optimization methods are good at dealing with low-dimensional expensive black-box problems and most multi-objective genetic algorithms lack convergence guarantees. For ES-based methods, such as ES, which optimizes only the mean and the variance is utilized purely to estimate the gradient. Those methods can not handle ill-conditional problems, as observed in our experiments on three synthetic problems. ES-based methods like CMA-ES, which optimizes mean and variance, are primarily designed for single-objective black-box optimization problems and may lack adaptations for handling MOO problems, while our methods are proposed for black-box MOO. We added these discussions in Section 3.1 of the updated version.
>
> ---
>
> >Q2. "Lack of comparison with more recent state-of-the-art methods. The authors only compare their algorithm with the ES-based method.''
>
> **A2**: Thanks for your suggestion. To improve the experimental section, we conducted additional experiments on two recently proposed methods: a random search method BES [1] and an evolution strategy method MMES [2]. The results of these two methods on both synthetic and black-box multi-task learning problems are put in Appendix F of the updated version. The results consistently demonstrate the effectiveness of the proposed ASMG method. The following tables show part of the result of these two baseline methods on the black-box multi-task learning settings. The first and the second tables show the experimental results on Office-31 and Office-home datasets, respectively.
>
> \begin{array}{ccccc}
>     \hline
>     \text{Method} & \text{A} & \text{D} & \text{W} & \text{Avg} \newline
>     \hline
>     \text{BES (d=256)} &  72.65 & 80.60 & 73.52 & 75.59\newline
>     \text{MMES (d=256)} & 75.90 & 83.33 & 76.67 & 78.63\newline
>     \text{ASMG (d=256)} & 77.83 & 86.61 & 80.56 & \textbf{81.67}\newline
>     \hline
>     \text{BES (d=512)} & 75.73 & 82.51 & 74.81 & 77.68 \newline
>     \text{MMES (d=512)} & 76.01 & 84.70 & 77.22 & 79.31\newline
>     \text{ASMG (d=512)} & 78.63 & 87.43 & 78.33 & \textbf{81.47}\newline
>     \hline
>     \text{BES (d=1024)} & 72.14 & 79.51 & 71.67 & 74.44\newline
>     \text{MMES (d=1024)} & 77.09 & 81.42 & 75.74 & 78.09\newline
>     \text{ASMG (d=1024)} & 76.30 & 87.70 & 80.19 & \textbf{81.40}\newline
>     \hline
> \end{array}
>
> \begin{array}{cccccc}
>     \hline
>     \text{Method} & \text{Ar} & \text{Cl} & \text{Pr} & \text{Rw} & \text{Avg} \newline
>     \hline
>     \text{BES (d=256)} & 68.94 & 45.97 & 81.53 & 79.42 & 68.97\newline
>     \text{MMES (d=256)} & 71.85 & 49.26 & 82.10 & 81.37 & 71.14\newline
>     \text{ASMG (d=256)} & 74.26 & 53.52 & 86.23 & 83.03 & \textbf{74.26}\newline
>     \hline
>     \text{BES (d=512)} & 69.39 & 48.18 & 82.94 & 80.29 & 70.20\newline
>     \text{MMES (d=512)} & 70.40 & 50.45 & 85.10 & 82.56 & 72.13\newline
>     \text{ASMG (d=512)} & 73.50 & 52.84 & 85.88 & 83.78 & \textbf{74.00}\newline
>     \hline
>     \text{BES (d=1024)} & 70.27 & 48.25 & 79.94 & 80.00 & 69.62\newline
>     \text{MMES (d=1024)} & 71.03 & 49.19 & 84.29 & 81.95 & 71.61\newline
>     \text{ASMG (d=1024)} & 73.18 & 51.82 & 85.84 & 83.21 & \textbf{73.51}\newline
>     \hline
> \end{array}
>
> As we discussed in the introduction section, the Bayesian optimization methods are not suitable for high-dimensional problems, i.e. $d=1024$, and multi-objective genetic algorithms explore the entire Pareto optimal set, which is computationally expensive for deep multi-task learning problems. Due to these two reasons and the difference in the algorithmic structure, we did not include those algorithms as baselines in the experiments.
>
> ---

---

> ### Author Response · Authors · 2023-11-18
> **Reply to Reviewer hUW7 (2/2)**
>
> >Q3. "Limited discussion of the algorithm's computational complexity and runtime.''
>
> **A3**: We now provide the computational complexity analysis for the proposed ASMG methods, and the corresponding discussion is put in Section 3.2 of the updated version.
> For the ASMG method, the computational cost of solving the weights in each iteration is negligible since the number of objectives is usually very small (i.e. $d>>m$). Therefore, the computational cost in each iteration is of order $\mathcal{O}(mNd)$. The results of real runtime (s) per iteration with various sample sizes $N$ in the black-box MTL settings when $d=512$ are shown in the following two tables. The first and the second tables show the experimental results on Office-31 ($m=3$) and Office-home ($m=4$) datasets, respectively. The experimental results are consistent with our analysis, and the real runtime of our method is similar to that of the CMA-ES and ES methods.
>
> \begin{array}{cccc}
>     \hline
>     \text{Method} & N=20 & N=40 & N=80 \newline
>     \hline
>     \text{ES} & 5.110 & 9.620 & 18.61\newline
>     \text{CMA-ES} & 5.236 & 9.680 & 18.75\newline
>     \text{ASMG} & 5.121 & 9.406 & 17.96\newline
>     \hline
> \end{array}
>
> \begin{array}{cccc}
>     \hline
>     \text{Method} & N=20 & N=40 & N=80 \newline
>     \hline
>     \text{ES} & 7.473 &  14.80 & 29.48\newline
>     \text{CMA-ES} & 7.647 & 14.60 & 28.88\newline
>     \text{ASMG} & 7.553 & 14.56 & 28.67\newline
>     \hline
> \end{array}
>
> ---
>
> >Q4. "How does the proposed algorithm compare to other black-box multi-objective optimization methods? Since only comparison with the ES-based method is discussed in the paper.''
>
> **A4**: Please see our reply to Q1 and Q2.
>
> ---
>
> >Q5. "What is the computational complexity and runtime of the proposed algorithm compared with other black-box multi-objective optimization methods?''
>
> **A5**: Please see our reply to Q3.
>
> ---
>
> **References**
>
> [1] Gao et al. Generalizing Gaussian Smoothing for Random Search. ICML, 2022.
>
> [2] He et al. MMES: Mixture Model-Based Evolution Strategy for Large-Scale Optimization. TEVC, 2021

---

> ### Author Response · Authors · 2023-11-21
> **We hope that our responses addressed your concerns**
>
> Dear reviewer hUW7,
>
> We'd like to express our gratitude once more for your constructive feedback, which lead to several updates that improve the understanding of our work. We are pleased that the reviewer found our work to be **novel and significant**. We've responded to each of your questions. We hope our responses adequately address your concerns. If you have any additional questions, we would be happy to follow up.
>
> Best wishes,
>
> Authors

---

> > ### Comment · Reviewer_hUW7 · 2023-11-23
> >
> > Thanks very much for your detailed clarification.

---

> > > ### Author Response · Authors · 2023-11-23
> > >
> > > Thanks for your reply. We are glad that our reply addressed your concerns.

---

### Author Response · Authors · 2023-11-18
**General Response**

Dear Reviewers,

We thank all the reviewers for their constructive and valuable comments.  We have answered each reviewer's questions separately. And we hope that our response addressed the reviewers' concerns.

We have revised the paper to address the comments and questions raised by reviewers, where the revision is highlighted in blue. The key updates include:

1. We added the discussion about the computational complexity of our method in Section 3.2, as suggested by Reviewers hUW7, ZfhH, and w58H.

2. We improved the experimental section, as suggested by Reviewers hUW7 and ZfhH. Additional experiments were conducted on two recently proposed black-box methods (i.e. BES and MMES), covering both synthetic problems and black-box multi-task learning (MTL) scenarios. The corresponding results are presented in Appendix F. We reported the standard deviation of the experimental results in the black-box MTL setting.

3. We added the related works about multi-objective optimization in the introduction section, including works about multi-task learning and learn-to-rank, as suggested by Reviewer J6Rt.

4. We added the discussion about the relevance of our method to previous works on MOO and stochastic gradient approximation in Section G, as suggested by Reviewer ZfhH.

Please check the updated version. Please let us know if there are any further concerns or questions.

Best,

The authors.

---

### Meta-Review · Area_Chair_h8DX · 2023-12-06

**Metareview:**

The paper proposes a ASMG algorithm for black-box multi-objective optimization with a theoretical guarantee. The authors have addressed the reviewers’ concerns and updated the paper based on the comments and suggestions.

**Justification For Why Not Higher Score:**

Weak support from the reviewers.

**Justification For Why Not Lower Score:**

All of the reviewers support the submission.

---

### Decision · Program_Chairs · 2024-01-16

Accept (poster)